# Conserved but mechanistically diverse piRNA defence against endogenous retroviruses in insects

Shashank Chary[1], Patricia E Carreira [2], Sarah Nicholas [3], Kathryn B McNamara[4], Ian A Cockburn[2], Karin Nordström [3], Therésa M Jones[4], Rosalyn Gloag [5], Alyson Ashe [6], Leon E Hugo[7,8,9] & Rippei Hayashi [1✉]

## Abstract

Defence systems against genetic mobile elements are highly adaptable, yet their long-term evolutionary stability remains unclear. To address this, we examined the conservation of Piwi-interacting RNA (piRNA)-mediated defence against *envelope*-carrying *gypsy* long terminal repeat (LTR) retrotransposons across insects beyond *Drosophila*. We show that *Aedes aegypti* (yellow fever mosquito) and *Anopheles stephensi* (Asian malaria mosquito), as well as *Tetragonula carbonaria* (stingless bees), *Acheta domesticus* (house cricket) and *Teleogryllus oceanicus* (Pacific field cricket), all produce piRNAs targeting *gypsy* elements in ovarian somatic cells—the same cellular niche where *Drosophila* mounts piRNA defence against *gypsy*—indicating a persistent arms race for more than 400 million years of insect evolution. Notably, in *Aedes aegypti*, ovarian somatic cells express the same piRNA clusters as other somatic tissues, where they are known to target RNA viruses—suggesting a shared origin of anti-viral and anti-retrotransposon defences. Furthermore, we observe lineage-specific differences in ovarian somatic piRNA biogenesis: slicing-independent phasing appears to dominate in dipterans, ping-pong amplification in bees, and slicing-dependent phasing in crickets. Together, these findings indicate that distinct piRNA pathways have independently evolved at different timepoints to silence the same class of retrotransposons in insect evolution.

**Keywords** Host Transposon Armsrace; piRNA-guided Gene Silencing; envelope-carrying retrotransposons
**Subject Categories** Evolution & Ecology; RNA Biology

## Introduction

Transposable elements or transposons are mobile sequences of DNA that can 'jump' around the genome, autonomously creating copies of themselves. Since individual insertions accumulate mutations and degrade over time, continuous transposition in the germline genome coupled to vertical transmission of new copies is vital to their long-term survival. Although most transposons replicate entirely within the cell in which they are expressed, some carry an *envelope* gene encoding membrane fusion proteins, enabling them to become infectious. A well-known example of such transposons occurs in *Drosophila*, where *envelope*-carrying *gypsy* long terminal repeat (LTR) retrotransposons, also called *gypsy* errantiviruses, are expressed in ovarian somatic cells and invade germline cells from the extracellular space before integrating into the germline genome (Song et al, 1994; Pelisson et al, 2002; Stefanov et al, 2012).

To mitigate the potential harm caused by their transpositions, *Drosophila* species acquired a dedicated defence system in the ovarian somatic cells based on the Piwi-interacting RNA (piRNA) pathway—an ancient and evolutionarily conserved guardian against transposons (Czech et al, 2018; Yamashiro and Siomi, 2018; Ozata et al, 2019). The piRNA pathway is a system of small RNAs produced from distinct genomic loci called piRNA clusters that are 10 to several hundred kilo bases long and abundantly contain transposon insertions. Small RNAs produced are antisense to the sequences of transposons and loaded onto the PIWI-clade Argonaute proteins, forming a piRNA-induced silencing complex. This complex then binds the transposon messenger RNAs by virtue of sequence complementarity and silences them through transcriptional and post-transcriptional mechanisms (Czech et al, 2018). The ovarian somatic piRNA pathway in *Drosophila* has diverged from its germline counterpart in multiple ways. Three PIWI proteins—Aubergine, AGO3 and Piwi—are expressed in the germline while only Piwi is expressed in the somatic cells (Brennecke et al, 2007). *piwi* refers to a specific gene of *Drosophila*, whose protein product is denoted as Piwi, whereas PIWI is a collective term for the clade of Argonaute proteins that associate with piRNAs. In the germline of *Drosophila*, piRNAs are generated from dual-stranded clusters containing transposon insertions in both sense and antisense orientations, whereas in somatic tissues, piRNAs arise from uni-stranded clusters, such as the one at the *flamenco* locus, which predominantly harbour transposon antisense insertions (Brennecke et al, 2007; Malone et al, 2009).

[1]The Shine-Dalgarno Centre for RNA Innovation, John Curtin School of Medical Research, The Australian National University, Acton, ACT 2601, Australia. [2]Division of Immunology and Infectious Disease, John Curtin School of Medical Research, The Australian National University, Acton, ACT 2601, Australia. [3]Flinders Health and Medical Research Institute, Flinders University, Bedford Park, SA 5042, Australia. [4]School of BioSciences, University of Melbourne, Parkville, VIC 3010, Australia. [5]School of Life and Environmental Sciences, The University of Sydney, Camperdown, NSW 2006, Australia. [6]Charles Perkins Centre, School of Life and Environmental Sciences, The University of Sydney, Camperdown, New 2006, Australia. [7]The Mosquito Control Laboratory, QIMR Berghofer, Herston, QLD 4006, Australia. [8]School of Biomedical Sciences, University of Queensland, St Lucia, QLD 4072, Australia. [9]Faculty of Health, Queensland University of Technology, Kelvin Grove, QLD 4059, Australia. ✉E-mail: rippei.hayashi@anu.edu.au

piRNA biogenesis mechanisms also differ between the germline and somatic cells in *Drosophila* reflecting the difference in the cluster architecture. In the germline, piRNA production is initiated by an endonucleolytic cleavage of a precursor RNA by Aubergine or AGO3—a process called 'slicing'. The 3' half of the cleaved RNA product is bound by another PIWI protein and undergoes a second processing step, which requires a cleavage by the endonuclease Zucchini/MitoPLD on the mitochondrial outer membrane or a 3'-to-5' exonucleolytic resection by Nibbler, to produce a mature piRNA (Han et al, 2015; Mohn et al, 2015; Hayashi et al, 2016). The ensuing secondary piRNA cleaves a precursor RNA from the opposite strand to generate the initial piRNA, thereby engaging in an amplification cycle called 'ping-pong' (Gunawardane et al, 2007). The production of the secondary piRNA also leads to a head-to-tail sequential production of piRNAs from the same strand by Zucchini/MitoPLD—a process called 'phasing' (Han et al, 2015; Mohn et al, 2015). piRNAs are further trimmed at the 3' end by the 3'-to-5' exonuclease Trimmer/Pnldc1 after the Zucchini/MitoPLD cleavage (Han et al, 2015; Izumi et al, 2016; Gainetdinov et al, 2018). While the identity of the piRNAs is dependent on the transposon landscape of the host, both ping-pong and phasing mechanisms are deeply conserved, from sponges to humans (Aravin et al, 2007; Grimson et al, 2008; Gainetdinov et al, 2018). In the ovarian somatic cells, Piwi—the sole PIWI protein expressed in these cells—is slicing-incompetent, consequently, the piRNA biogenesis proceeds independently of slicing (Brennecke et al, 2007). Instead, precursor transcripts from the uni-stranded clusters are recruited to Zucchini/MitoPLD by the DEAD-box RNA helicase Fs(1)Yb by a poorly understood mechanism to allow the phased piRNA production (Saito et al, 2010; Ishizu et al, 2019). The phased production of piRNAs from large uni-stranded clusters appears to be conserved in mosquitoes, where piRNAs are also expressed in non-gonadal somatic tissues to combat viral infections (Qu et al, 2023).

Both the *piwi* and *fs(1)yb* genes are duplicated derivatives of the Aubergine/Piwi subfamily of the PIWI genes and Tudor domain-containing protein family 12 (TDRD12) genes, respectively, which are uniquely present in *Drosophila* and closely related fly species (Handler et al, 2011; Lewis et al, 2016). This likely reflects the highly adaptive nature of the piRNA pathway, in which duplication and neofunctionalisation of its components frequently occur in evolution to counter the rapidly changing landscape of mobile elements (Miesen et al, 2015; Levine et al, 2016; Andersen et al, 2017; Lim et al, 2022). On the other hand, *envelope*-carrying *gypsy* elements—the main transposons active in the ovarian somatic cells of *Drosophila*, and therefore, silenced by piRNAs (Malone et al, 2009)—are widely present across metazoan phyla and particularly abundant in insects (Chary and Hayashi, 2025). This brings into question the conservation of the ovarian somatic piRNA pathway beyond *Drosophila*, and whether and how they have continued to silence the same elements over time.

# Results

## Intersection of ovarian and embryonic piRNA populations enables identification of somatic piRNA clusters

We and others previously demonstrated in *Drosophila* that genomic regions expressing significantly more piRNAs in the ovaries than embryos can be identified as ovarian somatic piRNA clusters because somatic piRNAs do not contribute to the early embryonic piRNA pool (Malone et al, 2009; Chary and Hayashi, 2023). To evaluate whether this approach is applicable to other insect species, we examined two other Brachycera flies—hoverflies (*Eristalis tenax*, Syrphidae) and Queensland fruit flies (*Bactrocera tryoni*, Tephritidae)—(Fig. 1A), both of which possess direct homologues of the *Drosophila piwi* gene (Lewis et al, 2016) and are therefore expected to express piRNAs in ovarian somatic cells. We collected embryos within a few hours of egg laying, along with ovaries from the same females that produced eggs (see Methods and Fig. 1B), and sequenced small RNAs that were resistant to periodate oxidation to enrich piRNAs. Genome-uniquely mapped reads were binned in 0.5 kb genomic tiles, and the coverage per tile was compared between ovarian and embryonic small RNA libraries. For both fly species, we identified genomic regions with piRNA production markedly enriched in the ovaries compared to embryos in two independent experiments (Figs. 1C and EV1A,D,D'). These genomic regions exhibit key features of the well-characterized ovarian somatic piRNA cluster *flamenco* in *Drosophila* (Brennecke et al, 2007): they predominantly generate piRNAs from a single strand and produce piRNAs that are antisense to *gypsy* LTR retrotransposons (Figs. 1D and EV1B,C,E,F). These findings indicate that intersecting ovarian and embryonic small RNA pools is an effective strategy for identifying ovarian somatic piRNA clusters.

## Mosquito ovarian somatic cells express piRNAs targeting *gypsy* and *BEL/pao* LTR retrotransposons as well as RNA viruses

Next, we investigated whether mosquitoes—most ancient relatives of flies within Diptera—express piRNAs targeting *gypsy* in ovarian somatic cells. *Aedes* mosquitoes have acquired a specialised piRNA pathway with an expanded number of PIWI family genes to defend against RNA viruses in non-gonadal somatic tissues (Morazzani et al, 2012; Miesen et al, 2015). Antiviral defence is mediated by cytoplasmic amplification of piRNAs from viral RNAs via the ping-pong mechanism, as well as by piRNAs derived from non-retroviral endogenous viral elements (nrEVEs), which are enriched in piRNA clusters expressed in non-gonadal somatic tissues (Miesen et al, 2015; Suzuki et al, 2020; Qu et al, 2023). Interestingly, we found that ovary-enriched piRNAs in *Ae. aegypti* are predominantly derived from these non-gonadal somatic piRNA clusters (Figs. 2A,B and EV2A). For example, cluster 3 (Fig. EV2B) expresses about 3% of all ovarian piRNAs—about a hundred times more than embryos (Fig. 2B). The expression of cluster 3 in the ovarian somatic cells was also confirmed by RNA Fluorescence in situ Hybridisation (FISH) (Fig. 2C',C"). The specificity of the FISH signal was verified by a FISH with a control probe set (Fig. EV2C). Analogous FISH experiments with control probe sets are presented in Expanded View Figures for all the other insects. Importantly, antisense insertions of *gypsy* are highly enriched in cluster 3 as well as in other somatic piRNA clusters, suggesting their role in targeting *gypsy* elements (Fig. 2B).

nrEVEs of antisense orientation are also enriched in these somatic piRNA clusters (Qu et al, 2023) (Figs. 2B and EV2D), suggesting that ovarian somatic cells are a battlefield against RNA viruses as well as *gypsy* retrotransposons (Janjoter et al, 2023). We

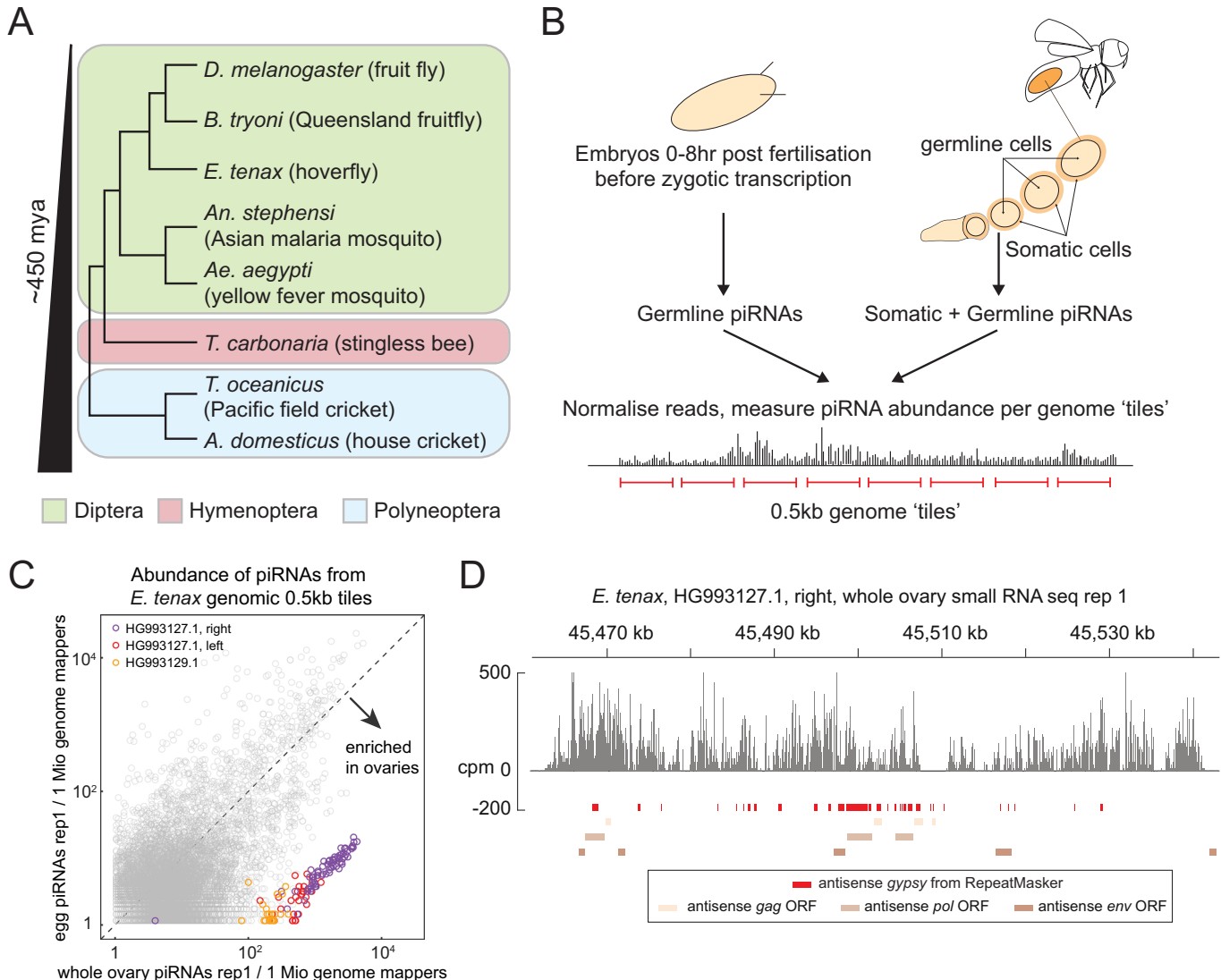

**Figure 1. Intersection of ovarian and embryonic small RNA libraries identifies somatic piRNA clusters.**

(A) Shown is a phylogenetic tree of the insect species used in this study, with dipteran, hymenopteran and polyneopteran species are grouped in different colours. (B) The strategy of somatic piRNA cluster identification is outlined. Ovarian and embryonic small RNA libraries were individually constructed and sequenced. Abundance of uniquely mapped piRNA reads was compared between the two libraries across the genomic 0.5 kb tiles. Tiles within the somatic piRNA clusters are expected to show greater coverage in the ovarian libraries. (C) A scatter plot showing the abundance of piRNAs from the ovaries (X axis) and the embryos (Y axis) that uniquely mapped to the individual 0.5 kb tiles of *E. tenax*. Tiles from newly identified clusters are highlighted in different colours. (D) Shown is the coverage of piRNA reads (> 22nt) in counts per million genome mappers (cpm) from the ovarian small RNA library of *E. tenax* that uniquely mapped to the piRNA cluster from contig HG993127.1. Reads that mapped to the plus- and minus-strands are coloured in dark and light grey, respectively. Coloured bars indicate *gypsy* insertions predicted by RepeatMasker and *gypsy* GAG, POL and ENV open reading frames predicted by tBLASTn.

further investigated this idea by searching for recent genomic insertions of RNA viruses producing piRNAs in ovarian somatic cells. We found piRNAs that matched to fragments of Liao ning virus from a search of 680 known mosquito viruses (Russo et al, 2019) (Figs. 2D and EV2E). Small RNA reads of approximately 25–29 nucleotides in length, displaying a 5' uridine bias—hallmarks of piRNAs—mapped to the Liao ning virus genome but not to the *Ae. aegypti* reference genome AaegL5.0 (Figs. 2E and EV2F,F'). This indicates that the Liao ning virus–derived fragments that generate piRNAs are polymorphic within the *Ae. aegypti* population: they are present in the Innisfail strain from Queensland, Australia, used

in this study, but absent from the Liverpool strain used for the AaegL5.0 genome assembly. Notably, piRNAs are only derived from the segment 5 but not from the other segments of the Liao ning virus (Fig. 2F). Interestingly, Liao ning virus is widespread in wild mosquito populations in Australia and has been shown to transmit vertically without infecting vertebrate animals (Prow et al, 2018). Although we did not observe typical response to strong viral infections where both sense and antisense strand produce piRNAs (Ma et al, 2021), our results indicate a recent acquisition of an nrEVE expressing piRNAs in ovarian somatic cells to provide immunity against RNA viruses in *Aedes* mosquitoes.

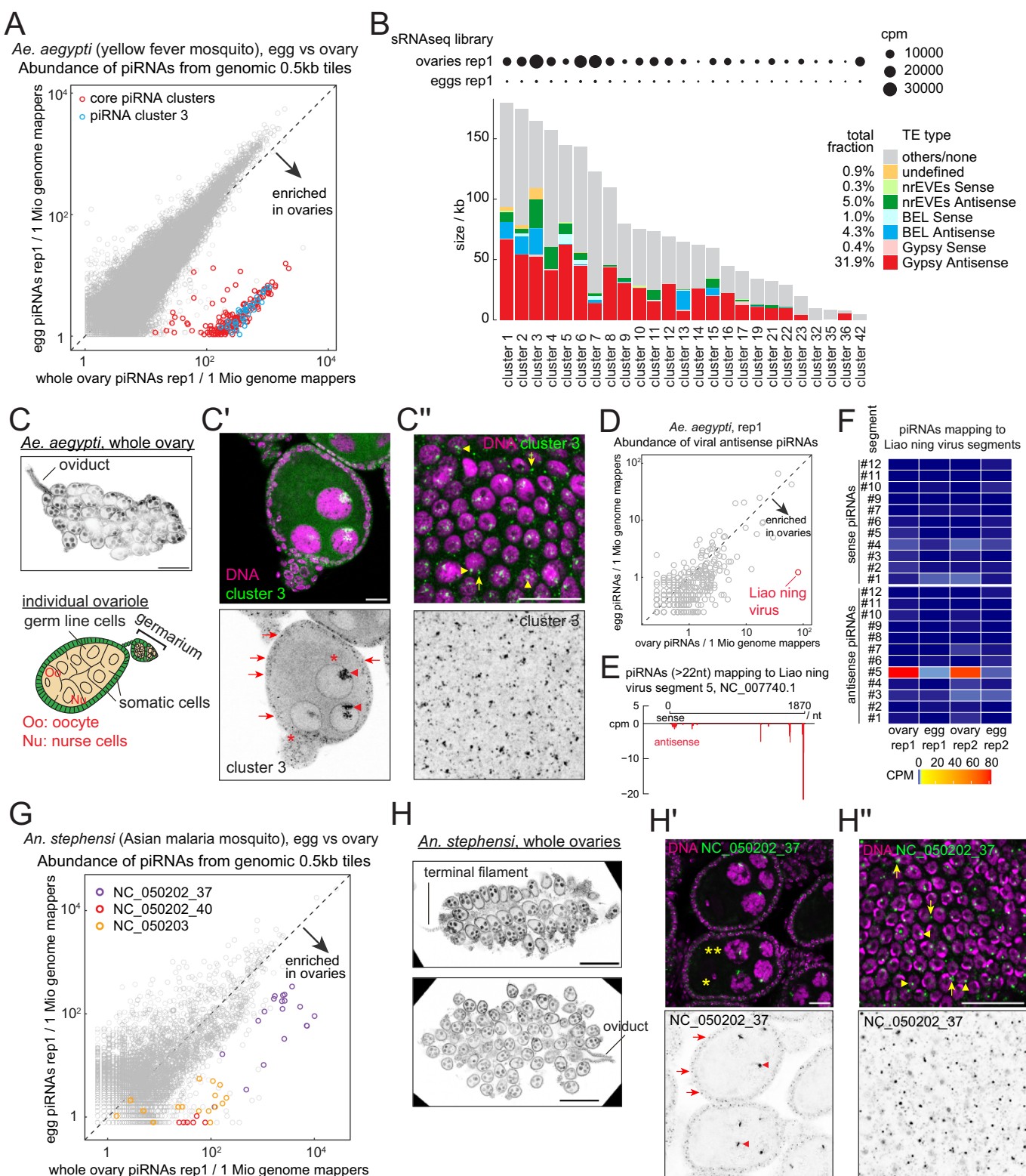

We next asked whether the ovarian somatic piRNA defence is also present in the malaria mosquito vector *Anopheles stephensi*, which is not known to be a prominent viral vector. A comparison of ovarian and embryonic small RNA libraries revealed a uni-stranded piRNA cluster with abundant antisense *gypsy* insertions, which is enriched in ovaries (Figs. 2G and EV2G,H). The expression of this piRNA cluster in the ovarian somatic cells was confirmed by RNA FISH (Figs. 2H',H"). Of the other two somatic

**Figure 2.  Mosquito ovarian somatic cells express piRNAs against *gypsy*, *BEL/pao* LTR retrotransposons and RNA viruses.**

(A) A scatter plot showing the abundance of piRNAs from the ovaries (X axis) and the embryos (Y axis) that uniquely mapped to the individual 0.5 kb tiles of *Ae. aegypti*. Tiles from the previously identified non-gonadal somatic piRNA clusters (Qu et al, 2023) and the piRNA cluster 3 are highlighted in different colours. (B) Shown are the size and composition of *gypsy* and *BEL* LTR retrotransposons—predicted by RepeatMasker, and nrEVEs within previously defined non-gonadal piRNA clusters in *Ae. aegypti* (Qu et al, 2023). Total fractions of each category across all clusters are shown. piRNA expression levels from ovarian and embryonic small RNAs (replicate 1) are also represented in circles. (C) A confocal image showing the whole ovary of *Ae. aegypti* stained by DAPI consisting of many ovarioles. The schematic of a single ovariole is shown below. (C', C") RNA FISH against transcripts from the piRNA cluster 3, showing its expression in the somatic cells (arrows) as well as in the germline nuclei (arrowheads). Peri-nuclear staining in the germline cells is likely a non-specific signal (asterisks, see Fig. EV2C). A bird's-eye view of the somatic epithelium of the FISH-staining is shown in (C"). Nuclear and cytoplasmic signals are indicated by arrowheads and arrows, respectively. Scale bars = 200 μm in (C) and 20 μm in (C') and (C"). (D) A scatter plot showing the abundance of piRNAs mapping to the antisense strands of annotated mosquito virus sequences, showing that piRNAs against Liao ning virus are enriched in the ovary library. (E) Shown is the 5' end coverage of piRNA reads (> 22nt) mapping to the Liao ning virus segment 5 from the ovarian small RNA library of *Ae. aegypti* in cpm. Sense and antisense reads are coloured in black and red, respectively. (F) Heatmap showing the abundance of piRNAs mapping to sense and antisense strands of individual Liao ning virus segments (1 to 12). CPM values are colour-coded in separate ranges: 0–2 (dark blue to sky blue) and 5–100 (yellow to red). (G) A scatter plot showing the abundance of piRNAs from the ovaries (X axis) and the embryos (Y axis) that uniquely mapped to the individual 0.5 kb tiles of *An. stephensi*. Tiles from newly identified clusters are highlighted in different colours. (H) Confocal images showing the whole ovary of *An. stephensi* stained by DAPI, showing similar structures as *Ae. aegypti* ovaries (C). (H', H") RNA FISH against transcripts from the piRNA cluster NC_050202_37, showing its expression in the somatic cells (arrows) as well as in the germline nuclei (arrowheads). Nurse cell and the oocyte nuclei are indicated by double and single asterisks, respectively. A bird's-eye view of the somatic epithelium of the FISH-staining is shown in (H"). Nuclear and cytoplasmic signals are indicated by arrowheads and arrows, respectively. Scale bars = 200 μm in (H) and 20 μm in (H') and (H"). Source data are available online for this figure.

piRNA clusters we identified, one cluster is located in a 3' UTR of a protein coding mRNA, and the other is found at an intergenic region enriched for antisense insertions of *BEL/pao* LTR retrotransposons (Fig. EV2H',H"). A previous study showed that some *BEL/pao* elements in mosquitoes carry *envelope* genes resembling the type III fusion glycoproteins from the Chuviridae viral family (Dezordi et al, 2020), implying a capability of infecting the germline cells. Interestingly, several somatic piRNA clusters in *Ae. aegypti* are also enriched for antisense *BEL/pao* insertions (Fig. 2B). These observations suggest that the piRNA pathway has a potential to silence a diverse range of RNA viruses in ovarian somatic cells, whether they are integrated in the genome or solely exist as RNA.

## Ovarian somatic piRNA pathway is conserved in non-dipteran insects

The ovarian gonadal niche may serve as a conserved host for retrotransposons/RNA viruses, which are silenced by piRNAs beyond dipterans. To investigate this possibility, we sequenced small RNAs from queen ovaries and freshly laid eggs of Australian stingless bees (*Tetragonula carbonaria*) to identify ovarian somatic piRNA clusters. Small RNA sequencing revealed an approximately 200 kb long piRNA cluster whose expression was enriched in the ovary libraries (Figs. 3A and EV3A). In addition to *gypsy*, antisense insertions of other transposon types such as LINE and *Copia* elements are found in this cluster (Fig. 3B). *Tetragonula* ovarioles resemble those of honeybee ovarioles, where the somatic cells can be distinguished from the oocytes that have larger nuclei (Martins and Bitondi 2016) (Fig. 3C,D). An RNA FISH experiment against the newly identified piRNA cluster confirmed its expression in the somatic cells (Fig. 3D,D'). Interestingly, although piRNAs deriving from one strand were dominant, they were also abundantly expressed from the other strand, suggesting that they were made by a different mechanism from ovarian somatic piRNAs in dipteran species (Fig. 3B and discussed later).

We further examined two cricket species—the house crickets (*Acheta domesticus*) and the Pacific field crickets (*Teleogryllus oceanicus*)—as representatives of hemimetabolous insects, from which holometabolous insects diverged more than 300 million

years ago (Nel et al, 2013). Analysis of ovarian and embryonic piRNAs revealed distinct piRNA clusters enriched in ovaries for both cricket species, containing antisense *gypsy* insertions (Figs. 4A–D and EV3C). An exception is cluster 8446_26 in *A. domesticus*, where only a few transposon insertions were found (Fig. EV3D). Basal insects such as crickets, cockroaches, and dragonflies contain panoistic ovaries, in which the oocyte, encapsulated by a layer of somatic cells, develops without supporting germline cells (Rumbo et al, 2023) (Figs. 4E and EV3E). An RNA FISH against cluster 8446_72 of *A. domesticus*, which carry *gypsy* insertions, showed its expression in ovarian somatic cells throughout the oogenesis with strong signals seen in apical somatic cells near the germline stem cells (Rumbo et al, 2023) (Fig. 4E,F).

In summary, we detected ovarian somatic cell-derived piRNAs across four dipteran insects (hoverflies, Queensland fruit fly, yellow fever mosquitoes, and Asian malaria mosquitoes), one hymenopteran (stingless bees) and two polyneopteran species (house crickets and Pacific field crickets) (Fig. EV4B). In all species, piRNAs mapping to *gypsy* elements were predominantly antisense and enriched in ovarian somatic cells compared to whole ovaries, consistent with a conserved and specialised defensive role against *gypsy* (Fig. EV4C). We further investigated the mobility of *gypsy* elements that are targeted by ovarian somatic piRNAs. In *Ae. aegypti*, we identified a *gypsy* insertion with GAG, POL and ENV ORFs within piRNA cluster 1 that shares more than 90% sequence identity with the previously described element *Aaeg_errantivirus_16* ((Chary and Hayashi, 2025), Fig. EV5A). A blastn search for *Aaeg_errantivirus_16* copies revealed that of 33 full-length insertions shared between the Liverpool strain (genome assembly: AaegL5.0) and the AEBAN2 isolate (IBAB_Aaeg_KPA_1.0), only 12 are present in the Rockefeller strain (CU_AaegROCK_1.0), suggesting ongoing mobilisation within *Ae. aegypti* populations. Similarly, we identified insertions with tandem GAG, POL and ENV ORFs in newly characterised ovarian somatic piRNA clusters of *An. stephensi*, *T. carbonaria*, *A. domesticus* and *T. oceanicus*, and found that their corresponding *gypsy* elements with more than 80% sequence similarity are present in multiple genomic copies (Fig. EV5B–E). These observations suggest that the arms race

between *gypsy* elements and ovarian somatic piRNA-mediated defence is ongoing, or was until recently, across a wide range of insect hosts.

## Distinct piRNA biogenesis mechanisms predominate in dipteran, bee and cricket ovarian somatic cells

The ovarian somatic piRNA pathway in *Drosophila* solely relies on the *piwi* gene, a Brachycera-specific derivative of the Aubergine/Piwi subfamily of PIWI clade Argonaute genes (Lewis et al, 2016). A phylogenetic analysis of annotated or newly identified Aubergine/Piwi proteins across insect orders including dipterans, silkworms, bees and crickets (see Methods) revealed that Aubergine/Piwi genes have duplicated multiple times in insect evolution (Fig. 5A). This and the wider conservation of ovarian piRNA pathway across insects prompted us to investigate its origin and potential mechanistic divergence.

An immuno-staining of Aubergine-Like in *An. stephensi*—a paralog of PIWI5 in *Ae. aegypti*, which dominantly receives piRNAs derived from somatic clusters (Qu et al, 2023)—showed cytoplasmic localisation both in the germline and somatic cells (Fig. 5B). The target specificity of the antibody was confirmed by western blotting (Fig. 5B'). Notably, this contrasts the nuclear localisation of Piwi proteins in germline and somatic cells of *Drosophila* ovaries (Brennecke et al, 2007). The peri-nuclear localisation of Aubergine-Like in the germline cells and focal localisation near the somatic nuclei are reminiscent of structures for piRNA biogenesis known as nuage and Yb body in *Drosophila*, respectively. This likely reflects its engagement in ping-pong and phased piRNA biogenesis, which were previously shown for PIWI5 (Miesen et al, 2015; Qu et al, 2023). Phased piRNA biogenesis—the dominant mechanism in the ovarian somatic cells of *Drosophila*—is mediated by the endonuclease Zucchini/MitoPLD, which preferentially cleaves RNA before uridine to simultaneously liberate the 5' and 3' ends of piRNAs (Han et al, 2015; Mohn et al, 2015) (Fig. 6A). The prevalence of phasing piRNA biogenesis can be assessed by measuring the frequency of uridine nucleotide at the immediate downstream nucleotides of piRNA 3' ends or the linkage between the 3' and 5' ends of piRNAs (Gainetdinov et al, 2018). Both signatures were present in piRNAs derived from ovarian somatic clusters of *Ae. aegypti* and *An. stephensi*, indicating that the phased piRNA biogenesis is predominant in ovarian somatic cells in dipterans (Figs. 6B,C,C' and EV6A,B). Notably, neither of the phasing signatures were present in *T. carbonaria*, *A. domesticus* and *T. oceanicus* (discussed later).

In contrast, piRNAs expressed from the *Tetragonula* ovarian somatic cluster showed a pattern characteristic of ping-pong piRNA biogenesis (Fig. 7A,B). In ping-pong piRNA biogenesis, two PIWI proteins cleave RNAs from plus- and minus-strands to generate each other's piRNAs (Brennecke et al, 2007; Gunawardane et al, 2007), which overlap exactly by 10 nucleotides at their 5' ends. Indeed, when the frequency of such overlaps between piRNAs from plus- and minus-strands was measured in individual 0.5 kb tiles across the cluster (ping-pong linkage analysis), most tiles showed a prominent peak at 10 nucleotides (Figs. 7C and EV6C,D), indicating a highly prevalent ping-pong biogenesis. Interestingly, piRNAs from plus- and minus-strands across the cluster showed distinct size distributions, with mean lengths of 27.4 and 25.3 nucleotides, respectively (Fig. EV6E). This suggests that the two populations are

selectively bound by different PIWI proteins—most likely the sole Aubergine/Piwi paralog and AGO3. Additionally, plus- and minus-strand piRNAs displayed strong enrichment (~ 90%) of uridine at their first and adenosine at their tenth nucleotide positions, respectively—another hallmark of the ping-pong cycle—further indicating that ping-pong piRNA biogenesis predominates in bee ovarian somatic cells (Fig. 7E and EV6F). Although the separation of somatic and germline-derived piRNAs is less distinct for *T. carbonaria* when comparing the ovarian and embryonic piRNA pools (Fig. 3A), it is unlikely that germline piRNAs are the primary source of the observed ping-pong signature as the cluster-derived piRNAs remained >10-fold enriched in the ovary libraries (Fig. EV4B). In contrast, the ovarian somatic clusters from *Ae. aegypti*, *An. stephensi*, *A. domesticus* and *T. oceanicus* predominantly produce piRNAs from single strand, therefore, they are unlikely to be produced by ping-pong (Fig. 7D).

A close inspection of piRNAs expressed in the cricket ovarian somatic cluster revealed occurrence of highly abundant piRNA peaks followed by trails of less abundant piRNAs (Figs. 8B and EV7A–C). This mode of piRNA production is characteristic of slicing-dependent phased piRNA biogenesis, which is known to be prominent in *Drosophila* germline and mouse Pachytene spermatocytes (Han et al, 2015; Mohn et al, 2015; Wu et al, 2020). There, a piRNA, which can be from a different genomic locus, slices an RNA to initiate phased piRNA biogenesis (Fig. 8A). The piRNA that triggers phased piRNA biogenesis and the piRNA made in response to the triggering event are called trigger and responder piRNAs, respectively. Interestingly, we found respective trigger piRNAs from different genomic loci for two of the most abundant responder piRNAs, each expressed in *A. domesticus* and *T. oceanicus* ovarian somatic clusters (Figs. 8B and EV7A–C). Responder and trigger piRNAs and their surrounding genomic regions showed extensive sequence complementarities with perfect overlaps between their most 5' ten nucleotides. Although the rule of piRNA-guided slicing and the trigger/responder pairing is incompletely understood, the region spanning positions g2 to g10 of piRNAs is generally expected to base-pair with the target sequence (Goh et al, 2015; Zhang et al, 2015). This linkage can be assessed using the so-called in-trans ping-pong analysis (Chary and Hayashi, 2023). In this approach, base-pairing between the 5' ends of piRNAs is examined, including those originating from distant genomic loci. When piRNAs from the ovarian somatic piRNA clusters and others that are enriched in the ovary libraries were analysed together, they showed a strong in-trans ping-pong linkage in both cricket species, further indicating the presence of slicing-dependent piRNA biogenesis in crickets (Fig. 8C). The linkage was absent in the ovarian somatic piRNA pool of *Ae. aegypti* and *An. stephensi*, suggesting that the phased piRNA biogenesis in these species is largely independent of slicing. Importantly, our results do not exclude the presence of slicing-dependent phasing particularly for *Ae. aegypti* where AGO3 homologue is known to be somatically expressed (Miesen et al, 2015).

Intriguingly, we found that the two most abundant responder and trigger piRNA pairs have identical sequences between the two cricket species (Fig. EV7D). Both piRNA clusters are devoid of protein-coding genes, and the regions around the two responder piRNAs in the cluster do not appear to contain transposon insertions (Fig. EV7E,F), suggesting that the sequence conservation

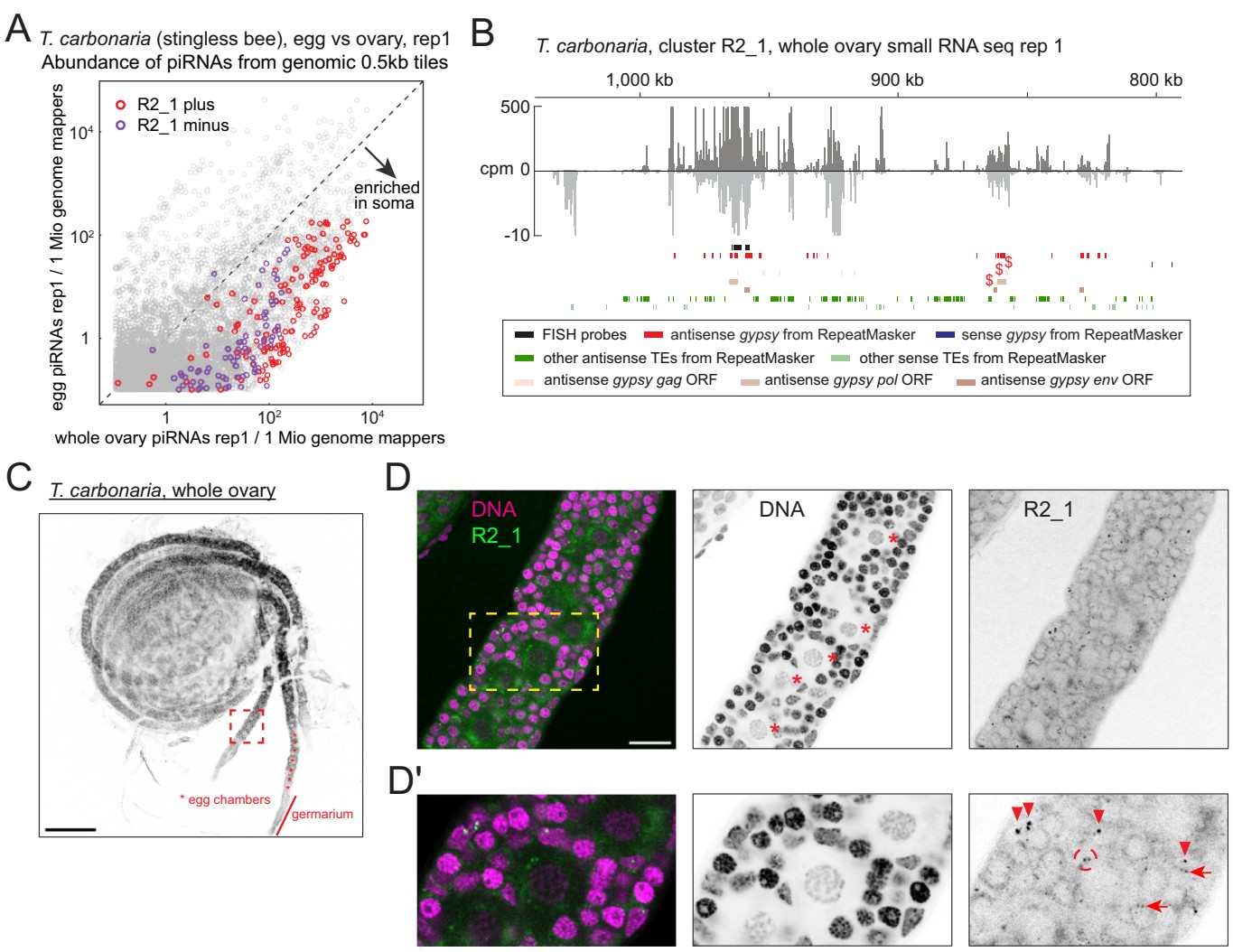

**Figure 3. *Tetragonula carbonaria* ovarian somatic cells express piRNAs against *gypsy* and other transposons.**

(A) A scatter plot showing the abundance of piRNAs from the ovaries (X axis) and the embryos (Y axis) that uniquely mapped to the individual 0.5 kb tiles of *T. carbonaria*. Plus- and minus-strand tiles from the newly identified piRNA cluster on the contig R2_1 are highlighted. (B) piRNA coverage plot across the cluster R2_1, shown in counts per million genome mappers (cpm) from replicate 1 of the ovarian small RNA library. Reads that mapped to the plus- and minus-strands are coloured in dark and light grey, respectively. Coloured bars indicate *gypsy* and other transposon insertions predicted by RepeatMasker, *gypsy* GAG, POL and ENV open reading frames predicted by tBLASTn, and the FISH probes. Antisense *gypsy* fragments that were examined for corresponding full-length insertions in Fig. EV5 are marked by dollars. (C) A confocal image showing the whole ovary of *T. carbonaria* stained by DAPI. The germarium and developing egg chambers are indicated. A close-up view of the dotted square is shown in (D). (D, D') A confocal image (D) and its close-up (D') of RNA FISH against transcripts from the plus-strand of the cluster R2_1, showing its expression in the somatic cells. Nuclear and cytoplasmic signals in the somatic cells are marked by arrowheads and arrows, respectively, and the signal in the germline cells is marked by a circle. Asterisks indicate the germline nuclei. All the other nuclei are of somatic cells. Scale bars = 200 μm in (C) and 20 μm in (D) and (D'). Source data are available online for this figure.

is unlikely due to their potential roles in mRNA processing or translation or transposon silencing. Although piRNA expression data are lacking, highly homologous genomic regions are present in other Gryllidae cricket species across multiple genera, suggesting over 100 million years of conservation (Fig. EV7D,D',D").

These observations collectively demonstrate that mechanistically distinct piRNA pathways have evolved in insects to target the same class of retrotransposons—*gypsy*—while also adapting to serve additional roles, such as silencing other transposons, RNA viruses, and potentially engaging in functions independent of transposon silencing (Fig. 8D).

## Discussion

Ovarian somatic piRNA pathway has long been considered to be uniquely acquired in *Drosophila* and related dipteran insects for two main reasons. First, the pathway depends on factors that are uniquely duplicated in *Drosophila* or closely related species—for example, the *piwi* gene, which is specifically acquired in Brachycera flies, and the paralog of the DEAD-box RNA helicase TDRD12, *fs(1)yb*, which is only present in *Drosophila*. Second, it has been proposed that *gypsy* errantiviruses—the main transposons active in ovarian somatic cells in *Drosophila*—acquired the *envelope* gene

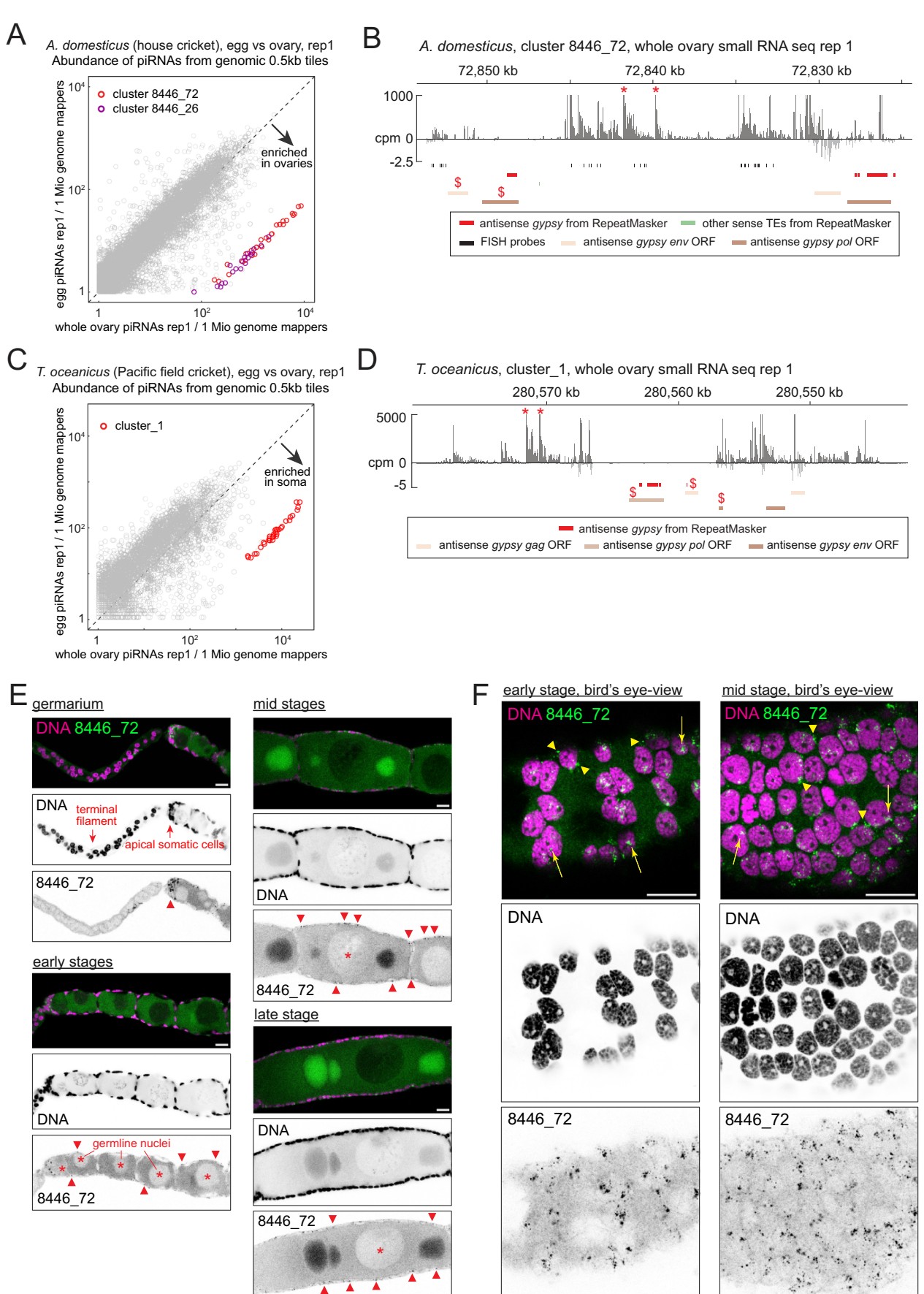

**Figure (A)** *A. domesticus* (house cricket), egg vs ovary, rep1. Abundance of piRNAs from genomic 0.5kb tiles.
- cluster 8446_72
- cluster 8446_26
- enriched in ovaries

**Figure (B)** *A. domesticus*, cluster 8446_72, whole ovary small RNA seq rep 1
- antisense *gypsy* from RepeatMasker
- other sense TEs from RepeatMasker
- FISH probes
- antisense *gypsy env* ORF
- antisense *gypsy pol* ORF

**Figure (C)** *T. oceanicus* (Pacific field cricket), egg vs ovary, rep1. Abundance of piRNAs from genomic 0.5kb tiles.
- cluster_1
- enriched in soma

**Figure (D)** *T. oceanicus*, cluster_1, whole ovary small RNA seq rep 1
- antisense *gypsy* from RepeatMasker
- antisense *gypsy gag* ORF
- antisense *gypsy pol* ORF
- antisense *gypsy env* ORF

**Figure (E)**
- germarium: DNA 8446_72, terminal filament, apical somatic cells
- early stages: DNA, germline nuclei, 8446_72
- mid stages: DNA, 8446_72
- late stage: DNA, 8446_72

**Figure (F)**
- early stage, bird's eye-view: DNA 8446_72, DNA, 8446_72
- mid stage, bird's eye-view: DNA 8446_72, DNA, 8446_72

**Figure 4. The ovarian somatic piRNA pathway is conserved in hemimetabolous crickets.**

(A, C) Scatter plots showing the abundance of piRNAs from the ovaries (X axis) and the embryos (Y axis) that uniquely mapped to the individual 0.5 kb tiles of *Acheta domesticus* in (A) and *Teleogryllus oceanicus* in (B). Tiles from newly identified clusters are highlighted in different colours. (B, D) piRNA coverage plots across the somatic piRNA cluster 8446_72 of *A. domesticus* in (B) and cluster 1 of *T. oceanicus* in (D), shown in counts per million genome mappers (cpm) from replicate 1 of the ovarian small RNA libraries from the respective species. Reads that mapped to the plus- and minus-strands are coloured in dark and light grey, respectively. Coloured bars indicate insertions of *gypsy* and other transposons predicted by RepeatMasker, *gypsy* GAG, POL and ENV open reading frames predicted by tBLASTn and the FISH probes. The two responder piRNAs discussed in Fig. 6 are indicated by asterisks. Antisense *gypsy* fragments that were examined for corresponding full-length insertions in Fig. EV5 are marked by dollars. (E, F) Side (E) and bird's-eye views (F) of the confocal images of RNA FISH against transcripts from the cluster 8446_72 of *A. domesticus* egg chambers, showing its expression in the somatic cells. In (E), FISH signals in the somatic cells are indicated by arrowheads and germline nuclei are marked by asterisks. Nuclear and cytoplasmic FISH signals are indicated by arrowheads and arrows, respectively, in (F). All nuclei that are visible are of the somatic origin in (F). Scale bars = 200 μm in (D) and 20 μm in (E) and (F). Source data are available online for this figure.

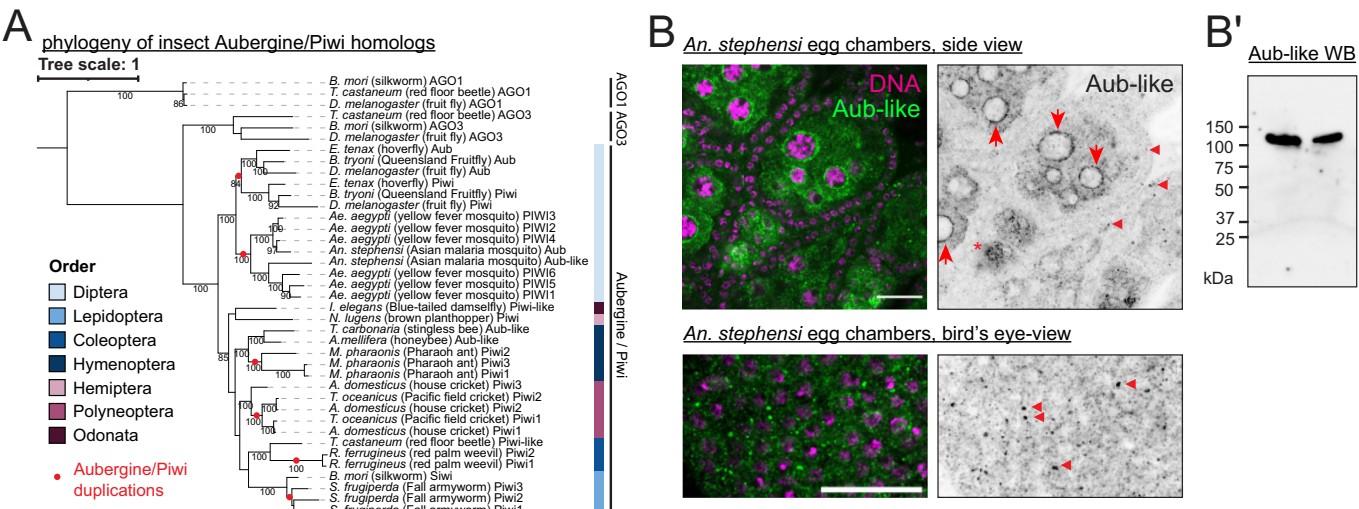

**Figure 5. Diversification of insect Aubergine/Piwi proteins including Aubergine-like in *Anopheles stephensi*.**

(A) Phylogenetic tree of Aubergine/PIWI homologues from various insects, with AGO1 and AGO3 from *D. melanogaster*, *B. mori* and *T. castaneum* included as known outgroups. Bootstrap values greater than 80 and the points of duplication are shown. (B) Immuno-fluorescent staining of Aubergine-Like in *An. stephensi* egg chambers, showing its peri-nuclear localisation in both somatic (arrowheads) and germline cells (arrows). Scale bars = 20 μm. (B') Western blot against Aubergine-Like on ovary lysates of *An. stephensi*. Two replicates were loaded on the membrane, showing the specific detection of the protein at about 110 kDa. Source data are available online for this figure.

from Baculoviruses, whose host range is limited to Dipterans, Lepidopterans, and Hymenopterans (Malik et al, 2000; Rohrmann and Karplus, 2001). Our recent works and this study challenge these views in multiple ways. First, the *fs(1)yb* gene has been lost in some *Drosophila* species while efficiently producing piRNAs against *gypsy*, suggesting that alternative pathways exist, even in *Drosophila*, for selectively producing transposon-targeting piRNAs (Chary and Hayashi, 2023). Second, *envelope*-carrying *gypsy* elements are widespread in metazoans, including animals outside Insecta (Chary and Hayashi, 2025). Lastly, we showed that ovarian somatic cells of a diverse range of insect species abundantly express piRNAs antisense to *gypsy* elements. These observations indicate that the conflict between *envelope*-carrying *gypsy* elements and the piRNA pathway has existed since the beginning of insect evolution or likely even before.

Targets of ovarian somatic piRNAs are likely not limited to *gypsy*. *Aedes* mosquitoes express piRNAs against at least three distinct classes of mobile elements—*envelope*-carrying *gypsy* and *BEL/pao* LTR retrotransposons, and RNA viruses (Figs. 2 and EV2). These different mobile elements may occupy distinct cell types

within the ovarian somatic cell niche, as recently demonstrated for *gypsy* elements in *D. melanogaster* (Senti et al, 2025). In that case, piRNAs targeting them also show differential expression. Similarly, in mosquitoes, insertions producing piRNAs against these elements are differentially distributed among piRNA clusters, suggesting a similar mechanism (Fig. 2). On the other hand, co-expression of LTR retrotransposons and RNA viruses in the same cells may benefit the host organism by increasing the chance of integrating the viral fragments into the genome via reverse transcription, thereby conferring a long-term immunity (Goic et al, 2016; Tassetto et al, 2019; Dezordi et al, 2020). Unlike dipteran and bee ovarian somatic piRNAs whose function is likely to target transposons and viruses, the function of the highly conserved cricket piRNAs remains elusive (Figs. 8 and EV7). They are unlikely targeting mis-annotated transposons or viruses given the rapid evolution of these elements and the likely chance of them acquiring mutations to evade silencing. They may have instead acquired a developmental role as shown for mouse pachytene piRNAs (Wu et al, 2020) or the piRNA involved in sex determination in silkworms (Kiuchi et al, 2014). Alternatively, the strong coupling of

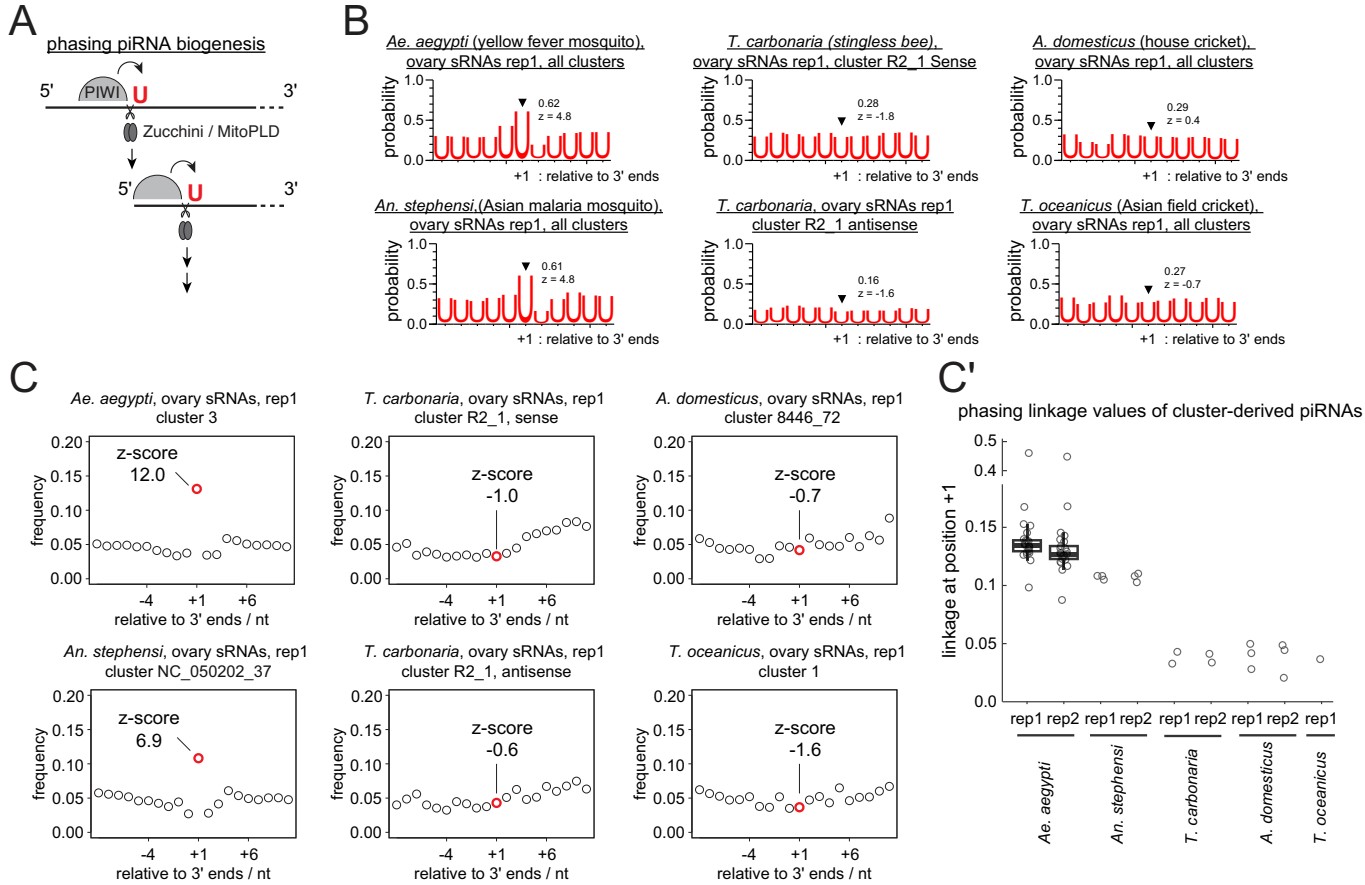

**Figure 6. Phased piRNA biogenesis predominates in mosquito ovarian somatic cells.**

(A) 'phasing' piRNA biogenesis is explained. piRNAs are produced by the endonuclease Zucchini/MitoPLD, which preferentially cleaves one nucleotide upstream of a uridine. (B) Shown are the frequencies of uridines found at positions relative to the 3' ends of piRNAs mapping to the ovarian somatic piRNA clusters of individual insect species. Frequencies at the linkage position (+1) and the z-scores are shown. piRNAs mapping to sense and antisense strands of cluster R2_1 were separately analysed for *T. carbonaria*. Ovarian small RNA library reads were analysed for all species. (C) Frequency plots of the distance between piRNA 3' and 5' ends from representative ovarian somatic clusters are shown for individual species. The Z-scores of the linkage distance +1, in which piRNA 5' ends are found immediately after piRNA 3' ends, are shown. (C') Plot showing the linkage values of phasing (+1) per species. Circles represent values from individual clusters. For *Ae. aegypti* replicates, the centre lines indicate the medians; box limits represent the interquartile range (IQR); whiskers extend to 1.5× IQR. N = 25 for *Ae. aegypti*, N = 3 for *An. stephensi* and *A. domesticus*, N = 2 for *T. carbonaria* and N = 1 for *T. oceanicus*.

the trigger and responder piRNAs, but not their sequences themselves, may need to be preserved. For example, frequent triggering events preferentially bring RNA close to the mitochondrial surface where the endonuclease Zucchini/MitoPLD resides, thereby facilitating the phased piRNA production (Ge et al, 2019). Targeted genetic manipulation of the trigger/responder piRNAs in the cricket genomes, and a survey of ovarian somatic piRNAs in a wider range of Orthoptera insects, will likely answer these questions.

We found that the ovarian somatic piRNA pathway in dipterans, bees and crickets each rely on distinct piRNA biogenesis mechanisms: phasing—devoid of slicing signature—for dipterans, ping-pong for bees, and slicing-dependent phasing for crickets (Figs. 6–8). Although signatures of phasing biogenesis were absent in cricket piRNAs (Fig. 6), the pattern of piRNA production suggests that Zucchini/MitoPLD continues to make piRNAs after the initial cleavage (Fig. 8). This apparent discrepancy is possibly

explained by exonucleolytic trimming of piRNA 3' ends, a maturation mechanism known in *Drosophila*, silkworm and mice (Han et al, 2015; Hayashi et al, 2016; Izumi et al, 2016; Gainetdinov et al, 2018). Both downstream uridine enrichment and a detectable 3' to 5' end linkage would only become evident if the relevant, yet unidentified, exonucleases were genetically depleted (Han et al, 2015; Gainetdinov et al, 2018). Likewise, exonucleolytic 3' end maturation and downstream phasing cannot be excluded in bees. In contrast, ovarian somatic piRNAs in dipterans are unlikely to arise through slicing, because piRNA 5' ends are protected by PIWI proteins once generated and are not known to undergo further processing. A question arises as to why and how different insect species acquired mechanistically different defence systems while targeting the same class of transposons. Once an efficient defence is established, there should be an evolutionary pressure for keeping it rather than replacing it with a different one. One possibility is that *envelope*-carrying *gypsy* elements arose or invaded these insects for

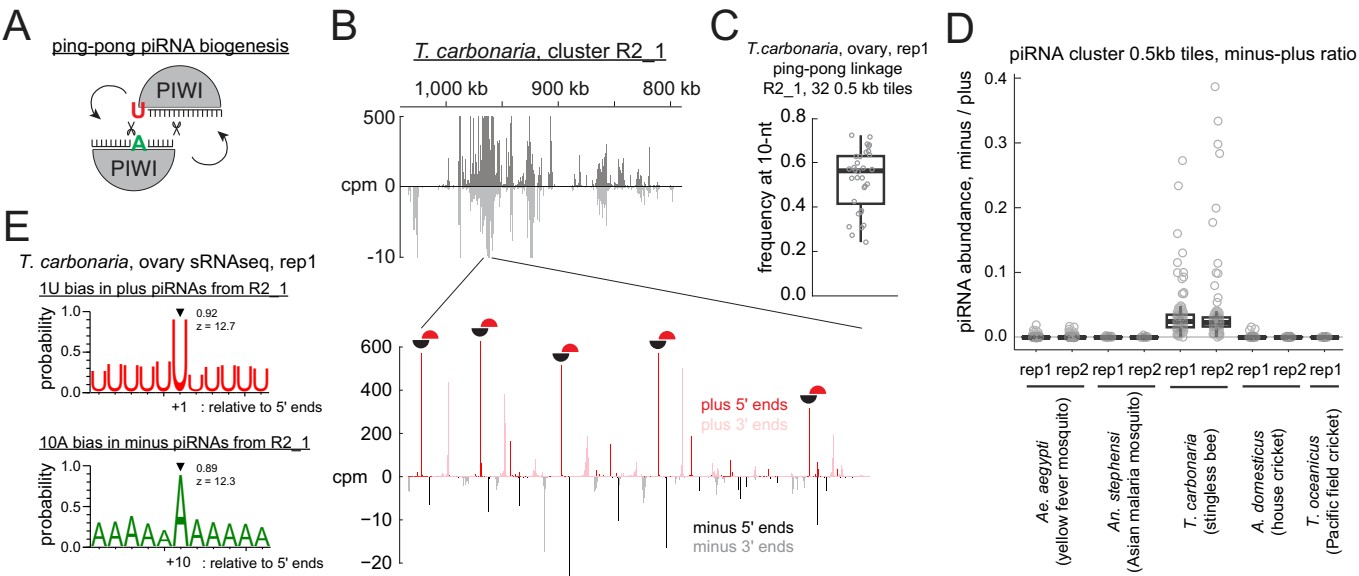

**Figure 7. Ping-pong piRNA biogenesis predominates in stingless bee ovarian somatic cells.**

(A) 'ping-pong' piRNA biogenesis is explained. Two piRNA-PIWI complexes amplify each other's production through slicing. (B) Close-up view of 963.0 - 962.5 kb region from the *T. carbonaria* cluster R2_1, showing the 5' and 3' end coverage of piRNAs from both strands. Prominent ping-pong pairs are marked by half-moons. (C) Boxplot showing frequencies at the 10-nt overlap from 0.5 kb tiles from cluster R2_1. $N = 32$ (D) Plot showing the relative abundance of piRNAs mapping to the minus strand over those mapping to the plus strand of the ovarian somatic piRNA clusters from respective species. $N = 179$ for *Ae. aegypti*, $N = 27$ for *An. stephensi*, $N = 104$ for *T. carbonaria*, $N = 48$ for *A. domesticus* and $N = 35$ for *T. oceanicus*. For boxplots in (C) and (D), circles represent values from individual 0.5 kb genomic tiles from the clusters. The centre lines indicate the medians; box limits represent the interquartile range (IQR); whiskers extend to 1.5× IQR. (E) Shown are the frequencies of Uridines and Adenosines found at positions around the 5' end and the tenth nucleotide of piRNAs mapping to the plus- and minus-strands of cluster R2_1. Frequencies at the linkage positions and the z-scores are shown.

the first time after they diverged, due to recombination of the *envelope* gene or horizontal transfer events. We consider this scenario unlikely, as the *envelope* genes associated with *gypsy* elements are highly ancient and also found in animals that diverged from insects in the early metazoan evolution (Chary and Hayashi, 2025). A more plausible explanation is that the requirement for piRNA-based defence may fluctuate over deep evolutionary time, driven by occasional losses and bursts of the transposons they target. In such cases, host organisms may repurpose components of the existing germline piRNA pathway when needed and subsequently lose them once the threat has diminished. Notably, *envelope*-carrying *gypsy* elements have been lost or absent in certain species. For example, entire echinoderm species appear to have lost them, despite their presence in sister deuterostome phyla such as Tunicata and Hemichordata (Chary and Hayashi, 2025). Likewise, honeybees (tribe Apini) possess an exceptionally small number of LTR retrotransposons, whereas these elements are abundant in bumblebees and stingless bees (Elsik et al, 2014; Sadd et al, 2015; de Souza Araujo et al, 2025). Absence of transposon activity in the ovarian somatic cells may lead to a loss of the defence system and eventually provide an opportunity for new *envelope*-carrying *gypsy* elements to come. Another possible explanation is that the ovarian somatic piRNA pathway may expand its toolkit in response to increased threats of mobile elements, which could allow the host to reconfigure its defence mechanism. For example, *Aedes* mosquitoes have undergone an extensive PIWI gene

duplication (Miesen et al, 2015) (Fig. 5), acquired a non-gonadal somatic piRNA pathway (Morazzani et al, 2012) and evolved multiple distinct piRNA-mediated transposon silencing mechanisms to counteract both retrotransposons and RNA viruses (Taşköprü et al, 2024). These innovations, often accompanied by expansion of PIWI and other Argonaute genes (Couvillion et al, 2009; Lewis et al, 2016; Seroussi et al, 2023) and the acquisition of both transcriptional and post-transcription gene silencing (Kuramochi-Miyagawa et al, 2008; Law and Jacobsen, 2010; Sienski et al, 2012), likely enable organisms to adapt to a diverse range of mobile elements and drive frequent reshaping of their defence systems. The expansion of molecular tools, followed by adaptive refinement, mirrors the "accordion model" frequently observed across diverse contexts of genetic conflict (McLaughlin and Malik, 2017).

In summary, we showed that the intersection of ovarian and embryonic piRNA pools enables robust identification of ovarian somatic piRNA clusters. We found that ovarian somatic cells across a wide range of insects express piRNAs targeting at least three distinct classes of retrotransposons and RNA viruses, most commonly the *gypsy* LTR retrotransposons. Despite this commonality, the piRNA pathway has repeatedly undergone reconfigurations in insect evolution. These findings highlight that the ovarian somatic cells is a previously underappreciated hotspot for evolutionary innovation in genetic mobile elements and the host defence systems.

# Methods

### Reagents and tools table

| Reagent/Resource | Reference or Source | Identifier or Catalog Number |
|---|---|---|
| **Experimental Models** | | |
| Hoverflies (*Eristalis tenax*) | Wild caught at Adelaide botanic gardens (Nicholas et al, 2018) | N/A |
| Queensland fruit flies (*Bactrocera tryoni*) | Macquarie University | N/A |
| Asian Malaria mosquitoes (*Anopheles stephensi*) | The Australian National University | N/A |
| Yellow fever mosquitoes (*Aedes aegypti*) | Innisfail strain, QIMR Berghofer Medical Research Institute | N/A |
| Australian stingless bees (*Tetragonula carbonaria*) | University of Sydney (Bueno et al, 2023) | N/A |
| House crickets (*Acheta domesticus*) | Amphibian research centre, Victoria | N/A |
| Pacific field crickets (*Teleogryllus oceanicus*) | Wild caught in Carnarvon, Western Australia (Simmons and Lovegrove, 2020) | N/A |
| **Recombinant DNA** | | |
| **Antibodies** | | |
| Anti *Anopheles stephensi* Aubergine-like | This study | N/A |
| **Oligonucleotides and other sequence-based reagents** | | |
| DNA oligos for RNA FISH | This study | N/A |
| Oligonucleotides for small RNA cloning | Chary and Hayashi (Chary and Hayashi, 2023) | N/A |
| **Chemicals, Enzymes and other reagents** | | |
| TRIzol | Thermo Fisher | 15596026 |
| RNase-free DNase I | New England Biolabs | M0303 |
| Triton X-100 | Sigma-Aldrich | X100-100ml |
| Goat anti-rabbit IgG conjugated to Alexa Fluor 568 | Abcam | ab175471 |
| cOmplete protease Inhibitors | Sigma-Aldrich | 4693132001 |
| 5-Propargylamino-ddUTP-Cy3 | Jena Biosciences | NU-1619-CY3 |
| 5-Propargylamino-ddUTP-Cy5 | Jena Biosciences | NU-1619-CY5 |
| Terminal Deoxynucleotidyl Transferase | Thermo Fisher | EP0161 |
| T4 RNA Ligase 2, K227Q | New England Biolabs | M0373L |
| T4 RNA Ligase 1 | New England Biolabs | M0204S |
| SuperScript II Reverse Transcriptase | Thermo Fisher | 18064014 |

| Reagent/Resource | Reference or Source | Identifier or Catalog Number |
|---|---|---|
| KAPA DNA polymerase | Sigma-Aldrich | KK3502 |
| **Software** | | |
| FASTX-toolkit | http://hannonlab.cshl.edu/fastx_toolkit/ | N/A |
| bowtie 1.2.3 | https://github.com/BenLangmead/bowtie/releases/tag/v1.2.3 | N/A |
| weblogo v3.7.8 | https://github.com/WebLogo/weblogo | N/A |
| RepeatMasker 4.1.0 | https://www.repeatmasker.org/RepeatMasker/ | N/A |
| ncbi-blast-2.9.0 | https://ftp.ncbi.nlm.nih.gov/blast/executables/blast+/2.9.0/ | N/A |
| bedtools 2.28.0 | https://github.com/arq5x/bedtools2/releases/tag/v2.28.0 | N/A |
| jellyfish 2.3.0 | https://github.com/gmarcais/Jellyfish/releases/tag/v2.3.0 | N/A |
| hhpred | https://toolkit.tuebingen.mpg.de/tools/hhpred | N/A |
| mafft 7.505 | https://mafft.cbrc.jp/alignment/software/source.html | N/A |
| iqtree-1.6.12 | http://www.iqtree.org/release/v1.6.12 | N/A |
| R 4.3.1 | https://cran.csiro.au/ | N/A |
| **Other** | | |

## Insect husbandry, egg and ovary collection

Hoverflies (*E. tenax*) were reared from wild-caught females, caught under permit from the Adelaide Botanic Gardens, as previously described (Nicholas et al, 2018). We collected >100 eggs per replicate from either the wild-caught or the laboratory-reared females within an hour of laying. Three to four pairs of ovaries from 3-month-old laboratory-reared females were collected per replicate.

Queensland fruit flies (*B. tryoni*) are continuously bred in a controlled environmental laboratory in Macquarie University. Pupae were brought and reared in the Australian National University. Adults were fed with hydrolysed yeast and sugar until females become mature and lay eggs within a week of eclosion. For each replicate, about 200 eggs laid within two hours and five to ten ovaries were pooled.

*An. stephensi* and *Ae. aegypti* (Innisfail strain) were bred in insectary facilities at the Australian National University and QIMR Berghofer Medical Research Institute, respectively. *Anopheles* and *Aedes* female mosquitoes aged approximately four days were provided the opportunity to blood feed on an anasthesied mouse or the arm of a volunteer under ANU AEC protocol number 2022/36 and QIMR Berghofer Human Research Ethics approval P2273, respectively. Three days later, a container of water lined with filter paper was added to the cage for females to lay eggs. The eggs were collected from the paper within four hours of being laid, after which the females were dissected to isolate ovaries. For each replicate, about 100 eggs and 20 pairs of ovaries were pooled.

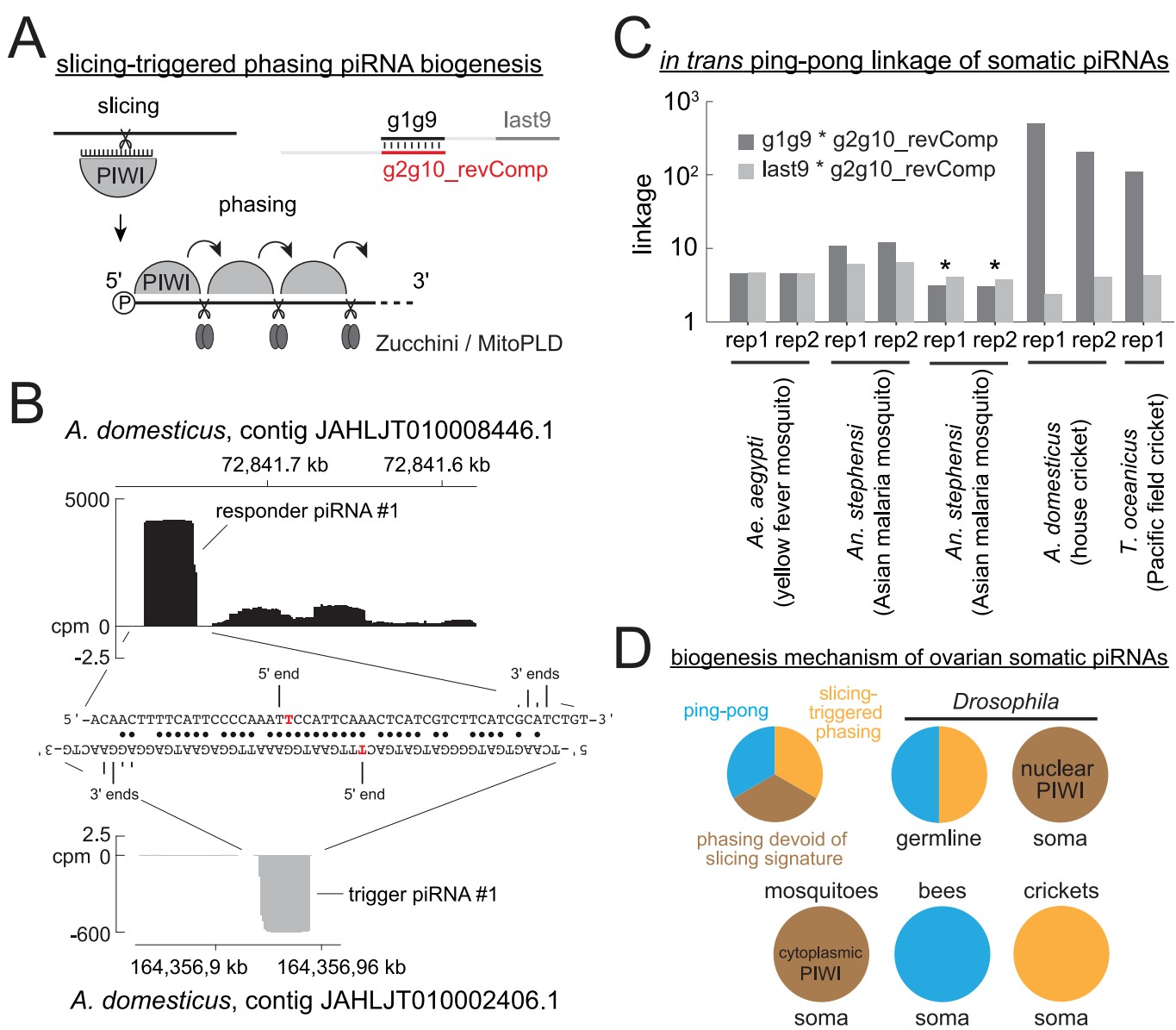

**Figure 8. Slicing-triggered phasing piRNA biogenesis occurs in cricket ovarian somatic cells.**

(A) 'slicing-triggered' phasing piRNA biogenesis is explained. (B) Sequences and the cpm coverage of genomic regions at responder and trigger piRNA pairs #1 from *A. domesticus*. The 5′ and 3′ end positions are indicated with frequencies represented by heights for the 3′ ends. Watson-Crick base pairs are marked by circles. (C) In-trans ping-pong linkage values of piRNAs uniquely mapped to the somatically enriched genomic tiles (> 7-fold for *An. stephensi* and >10-fold for the other species) are shown for two replicates of *Ae. aegypti*, *An. stephensi*, and *A. domesticus*, and one replicate of *T. oceanicus* ovarian small RNA libraries. The linkage values between piRNAs derived exclusively from the ovarian somatic clusters are also shown for *An. stephensi* (marked by asterisks). (D) piRNA biogenesis mechanisms that predominantly occur in *Drosophila*, mosquitoes, stingless bees and crickets are shown.

Colonies of the Australian stingless bees (*T. carbonaria*) were kept in The University of Sydney. Eggs were collected by opening hives and removing a disc of brood cells at the brood's advancing front with a scalpel. Brood cells on the outer edge of the disc and adjacent to open cells, which contained the most recently laid eggs— estimated age to be within 8 h, were opened under a stereo microscope and the egg removed from the surrounding food using a platinum wire. Queens were collected by harvesting pupal queen cells from colonies and allowing them to emerge in petri-dish micro-colonies (Bueno et al, 2023). Queens were snap frozen in liquid nitrogen at 5–7 days of age,

and subsequently dissected to isolate ovaries. For each replicate, approximately 100 eggs and 10 pairs of ovaries were pooled.

Pacific field crickets (*T. oceanicus*) were derived from a wild-caught population located in Carnarvon, Western Australia and were then maintained under standard conditions (Simmons and Lovegrove, 2020). House crickets (*A. domesticus*) were obtained from the Amphibian Research Centre, Victoria. Crickets were housed in containers with vermiculite and provided ad libitum access to water, fresh vegetable and cat chow. 30 to 50 eggs laid within 8 h, and one or two pairs of ovaries dissected from

approximately 2-week-old sexually mature females were collected per replicate.

Ovaries and eggs were homogenised in TRIzol to obtain the total RNA. The total RNA was further treated with RNase-free DNase I (NEB, M0303) before proceeding with the small RNA cloning.

## Immuno-fluorescence staining of ovaries

Rabbit polyclonal antibodies against *A. stephensi* Aubergine-like (XP_035903683.1) were generated by Genscript using a peptide RRANAPSKQGASGTc where an additional single cysteine residue for the cross-linking purpose is indicated in lowercase. Purified IgG were reconstituted in PBS (0.5 µg/ml) before use. The ovaries were freshly dissected from 1-week-old females in PBS and fixed in PBS containing 4% formaldehyde for 10 min at room temperature. Fixed ovaries were permeabilised in PBS containing 0.5% v/v Triton-X for 30 min, washed in PBS containing 0.1% Triton-X (PBS-Tx) several times before blocking in PBS-Tx containing 0.05% w/v BSA for 30 min. The primary antibody incubation was conducted in the blocking solution at 4 °C overnight with antibodies in 1:500 dilution. Goat anti-rabbit IgG conjugated to Alexa Fluoro 568 (Abcam) were used as secondary antibodies and the confocal images were taken on a Zeiss LSM-800. DAPI was used to visualise DNA. Images were processed by Fiji.

## Western blotting

5 to 10 µl of freshly dissected ovaries of *An. stephensi* was collected in cold PBS and snap-frozen. The ovaries were homogenised in 100 µl of the RIPA buffer (50 mM Tris-HCl pH 7.5, 150 mM NaCl, 1% Triton X-100, 0.1% SDS, 0.1% Na-deoxycholate, 1 mM EDTA, and 1× cOmplete protease Inhibitors (Roche)) on ice. The lysate was cleared by centrifugation, boiled in Laemmli buffer before loading onto the SDS-PAGE. The gel was transferred onto a Nitrocellulose membrane and incubated with rabbit polyclonal anti-Aubergine-like antibodies at a 1:2000 dilution. The HPR-conjugated secondary antibodies were used for the standard ECL detection.

## RNA in situ fluorescent hybridisation (FISH)

RNA FISH protocol was previously described (Chary and Hayashi, 2023). Briefly, short oligo DNAs for each target (see Dataset EV1) were pooled and labelled with 5-Propargylamino-ddUTP-Cy3 or -Cy5 (Jena Biosciences) using the Terminal Deoxynucleotidyl Transferase (Thermo Fisher). Freshly dissected ovaries from different insects were fixed in PBS containing 4% formaldehyde. The fixed ovaries were permeabilised overnight at 4 °C in 70% ethanol and washed twice by RNA FISH wash buffer (10% (v/w) formamide in 2x SSC). Subsequently, the ovaries were resuspended in 50 µL Hybridization Buffer (10% (v/w) dextran sulfate and 10% (v/w) formamide in 2× SSC) and incubated with 1 pmol of labelled oligo probes for overnight at 37 °C. The ovaries were then washed several times in RNA FISH Wash Buffer and stained by DAPI before mounting. Confocal images were taken on a Zeiss LSM 800 and processed by Fiji.

## Small RNA cloning

We generated small RNA libraries from 5 µg of oxidised total RNA as previously described (Chary and Hayashi 2023). Briefly, the size of 19 to 35 nt of RNA was selected from the total RNA by PAGE using radio-labelled 19mer spike (5'-CGUACGCGGGUUUAAACGA) and 35mer spike (5'-CUCAUCUUGGUCGUACGCGGAAUAGUUUAAA-CUGU). The size-selected RNA was precipitated, oxidised by sodium periodate, and size-selected for the second time by PAGE. The size-selected small RNAs were ligated to the 3' adaptor from IDT (5rApp/NNNNAGATCGGAAGAGCACACGTCT/3ddC where Ns are randomised) using the truncated T4 RNA Ligase 2, K227Q (NEB), followed by a third PAGE. Subsequently, the RNA was ligated to the 5' adaptor from IDT (ACACUCUUUCCCUACACGACGCUCUUCCGAUCUNNNN where Ns are randomised) using the T4 RNA Ligase 1 (NEB). Adaptor-ligated RNA was reverse-transcribed by SuperScript II and amplified by KAPA polymerase using the universal forward primer, Solexa_PCR-fw: (5'-AATGATACGGCGACCACCGAGATCTACAC TCTTTCCCTACACGACGCTCTTCCGATCT) and the barcode-containing reverse primer TruSeq_IDX: (5'-CAAGCAGAAGACGGCA-TACGAGAT__xxxxxx__GTGACTGGAGTTCAGACGTGTGCTCTTCC-GATCT where xxxxxx is the reverse-complemented barcode sequence). Amplified libraries were multiplexed and sequenced on HiSeq or NovaSeq platforms in the paired-end 150 bp mode by GENEWIZ/ Azenta.

## Processing and mapping of small RNA sequencing reads

The R1 sequencing reads were trimmed of the Illumina-adaptor sequence using the FASTX-Toolkit. The four random nucleotides at both ends of the read were further removed. The trimmed reads of 18 to 40 nt in size were first mapped to the infra-structural RNAs, including ribosomal RNAs, small nucleolar RNAs, small nuclear RNAs, microRNAs, and transfer RNAs using bowtie 1.2.3 allowing up to one mismatch. Sequences annotated in the following genomes were used: GCF_945859705.1_idEpiBalt1.1 from *Episyrphus balteatus* for hoverflies, GCF_016617805.1_CSIRO_BtryS06_freeze2 for Queens-land fruit flies, GCF_002204515.2_AaegL5.0 for *Ae. aegypti*, GCF_013141755.1_UCI_ANSTEP_V1.0 for *An. stephensi*, GCF_003254395.2_Amel_HAv3.1_genomic from *Ames mellifera* for *Te. carbonaria*, and GCF_023897955.1_iqSchGreg1.2 from *Schisto-cerca gregaria* for crickets. The remaining reads were used for all the downstream analyses. The trimmed and unfiltered reads were then mapped to the GCA_905231855.2_idEriTena2.2 assembly of the *E. tenax* genome, the GCF_016617805.1_CSIRO_BtryS06_freeze2 assembly of the *B. tryoni* genome, the GCF_002204515.2_AaegL5.0 assembly of the *Ae. aegypti* genome, the GCF_013141755.1_UCI_AN-STEP_V1.0 assembly of the *An. stephensi* genome, the GCA_032399595.1_TetCarb_2.0 assembly of the *T. carbonaria* genome, the GCA_031308135.1_ado_dt_MfTpo_fix assembly of the *A. domesticus* genome, and the GCA_964035755.1_iqTelOcea1 assembly of the *T. oceanicus* genome, using bowtie allowing up to one mismatch. The genome-unique mappers were used to visualise piRNAs expressed from the piRNA clusters. Additionally, small RNA reads were mapped to the annotated mosquito virus genomic sequences from the previous study (Russo et al, 2019) using bowtie allowing up to three mismatches. Nucleotide frequencies were analysed around key positions of piRNA reads (23nt or longer) mapped to the R2_1 piRNA cluster of the *T. carbonaria* genome, specifically at the first nucleotide of plus-strand reads and the tenth nucleotide of minus-strand reads for ping-pong biogenesis. Similarly, frequencies around the 3' end of piRNAs mapped to the piRNA clusters of *Ae. aegypti* and *An. stephensi* were examined for phasing biogenesis. Frequencies were calculated within an 11-nucleotide

window and visualized using weblogo v3.7.8. Z-scores were calculated as the deviation of the central nucleotide frequency from the mean, divided by the standard deviation of the frequencies.

## Identification of transposon insertions

We ran RepeatMasker 4.1.0 to predict transposon insertions in respective genome assemblies using the RepBaseRepeatMaskerEdition-20181026. We separately ran tBLASTn (ncbi-blast-2.9.0) to search for fragments of *gag*, *pol* and *envelope* Open Reading Frames of the *envelope*-containing *gypsy* elements using the bait sequences of 289 representative metazoan errantiviruses that were previously identified (Chary and Hayashi, 2025). The sequences of GAG, POL, ENVELOPE baits can be found in the git repository (https://github.com/RippeiHayashi/insect_gypsy). tBLASTn was run with default parameters, applying an E-value cutoff of ≤1e–10. Genomic regions that showed homology to the POL baits with a bit score of greater than 50 or to the GAG and ENV baits with a bit score of greater than 30 were individually pooled. Genomic regions that had hits from five or more baits were merged for their coordinates using bedtools/2.28.0. Regions that were larger than 1500 nucleotides for POL and 300 nucleotides for GAG or ENV were collected. Predicted transposon insertions are displayed in the genome browser tracks alongside the small RNA libraries. Additionally, predicted non-retroviral integrated RNA virus sequences—also known as non-retroviral endogenous viral elements (nrEVEs)—from the previous study of the *Ae. aegypti* genome (Russo et al, 2019) are included.

## Identification of genomic *gypsy* insertions that are targeted by ovarian somatic piRNAs

We used BLASTn to search for genomic *gypsy* insertions that are outside the ovarian somatic piRNA clusters and yet are highly homologous to the GAG, POL and ENV fragments found within the clusters. The following are the genomic coordinates of bait GAG/POL/ENV fragments for the BLASTn search: *An. stephensi* GAG (NC_050202.1:37718689-37719944( + )), POL (NC_050202.1:37719925-37723060(+)) and ENV (NC_050202.1:37723050-37723898( + )); *T. carbonaria* GAG (R2_1:857659-858503( + )), POL (R2_1:858371-861768(+)) and ENV (R2_1:861806-863155( + )); *T. oceanicus* GAG (CAXIVR010000001.1:280558500-280559500( + )), POL (CAXIVR010000001.1:280561098-280563731(+)) and ENV (CAXIVR010000001.1:280556653-280556965(+)). We used the full-length *gypsy* insertions *Aaeg_errantivirus_16* (Chary and Hayashi, 2025) and *Adom_errantivirus_1* (identified in this study: JAHLJT010002406.1:32971394-32982678(+)) as baits for *Ae. aegypti* and *A. domesticus*, respectively. Genomic copies of *Aaeg_errantivirus_16* were identified as insertions with >98% homology across >80% of the transposon sequence including both LTRs. Genomic copies of *Adom_errantivirus_1* were identified as insertions with >80% homology across >80% of the transposon sequence including both LTRs.

## Quantification of *gypsy* sense and antisense piRNAs

After filtering infra-structural RNA reads, we mapped the sequencing reads to the genome using bowtie with the "--all --best --strata" option and divided individual mapping instances by the number of mapping events per read, in order to evenly distribute multi-mappers across all repeats in the genome. We counted the number of reads mapping to individual *gypsy* insertions, including those annotated by RepeatMaster and tBLASTn (see above), using bedtools intersect. We then calculated the sum to determine the total abundance of *gypsy* sense and antisense piRNAs.

## Tile coverage analysis of the small RNA sequencing reads

We carried out genomic tile analyses to measure the abundance of piRNAs. We only considered reads that are 23 nt or longer as piRNAs. We first identified 0.5 kb tiles that are covered at least 85% by unique regions by mapping artificially made 25mers against the whole genome. We then counted the number of genome-unique reads that mapped to individual uniquely-mappable 0.5 kb tiles. Counts were normalised to the total number of genome-unique piRNA mappers. The coordinates of the ovarian somatic piRNA clusters used in this study can be found in the Dataset EV2.

## Linkage analysis of phasing piRNA biogenesis

We carried out the phasing linkage analysis of piRNAs using normalised genome all mappers as previously described (Gainetdinov et al, 2018). First, abundance of 5' and 3' ends of piRNA mappers were measured across the ovarian somatic piRNA clusters. Frequencies of the distance between each 3' end and 5' ends found in the window of 20nt centred on that 3' end were measured and weighted by the abundance of 3' end. Z-scores were calculated as the deviation of the frequency value at the linkage position from the mean frequency divided by the standard deviation of the frequencies.

## Ping-pong linkage analysis

We carried out the ping-pong linkage analysis of piRNAs (23 nt or longer) that were uniquely mapped to the piRNA cluster in the R2_1 contig of the *T. carbonaria* genome. Individual 0.5 kb tiles from the piRNA cluster with coverage exceeding 1000 and 10 counts per million genome-mapped reads on plus-strand and minus-strand strands, respectively, were assessed for ping-pong signature. Briefly, 5' and 3' ends of piRNA mappers were counted at each tile coordinates. Frequencies of the co-occurrence at the linkage position ( + 10 between 5' ends of plus and minus piRNAs) were measured in the window of 20nt across the tile and weighed by the abundance of piRNAs. Z-scores were calculated as the deviation of the frequency value at the linkage position from the mean frequency divided by the standard deviation of the frequencies.

## Linkage analysis of in-trans ping-pong

The linkage analysis was previously described (Chary and Hayashi, 2023). Briefly, individual genome-mapping piRNA reads were trimmed to the following three regions: the g1 to g9 position (g1g9) and the most 3' 9 nucleotides (last9) of the sense piRNA read; and the reverse-complemented sequence from the g2 to g10 position (g2g10_revComp) where the g1 is the 5' end of a piRNA and the g2

is the penultimate 5' end, and so on. The frequencies of every different 9mers from each category were counted using jellyfish 2.3.0 and normalised to one million genome-mappers. The sum of the frequencies was arbitrarily set to 1000. Products of the frequencies from the g1g9 and the g2g10_revComp were calculated for each 9mer and summed up to yield the in-trans ping-pong linkage value. The linkage value increases when two piRNAs form an in-trans ping-pong pair and the same sequence occurs at higher frequencies both in the g1g9 and the g2g10_revComp. We do not expect to observe any linkage between the last9 and the g2g10_revComp, therefore, the products between them were considered a genomic background. For *Ae. aegypti*, *A. domesticus* and *T. oceanicus*, we used piRNA reads that uniquely mapped to genomic 0.5 kb tiles showing >10-fold enrichment in the whole ovary small RNA libraries relative to embryonic libraries. For *An. stephensi*, tiles with >7-fold enrichment in the ovary libraries were used. In *An. stephensi*, we also analysed the linkage specifically among piRNAs originating from the ovarian somatic clusters. During this analysis, we found an abundant piRNA, TTTCTAT-TAAATATGTTGCAATCAACCG and its ping-pong partner, TTAATAGAAATTACAGTGGTTGGTATCTT, both mapping to a tile spanning positions 351,125,500–351,126,000 on chromosome NC_035109.1 of *Ae. aegypti*. Further analysis revealed that piRNAs beginning with "TTTCTATTAA" and their corresponding ping-pong pairs beginning with "TTAATAGAAA", which mapped to the genome at multiple sites, were highly expressed both in the whole ovaries and embryos. Because canonical ping-pong piRNA pairs can artificially inflate in-trans ping-pong linkage signals, we excluded this tile from the analysis.

### Phylogenetic analysis of Aubergine/Piwi homologues

The following sequences of the PIWI proteins were used for the anlaysis. FBpp0309202 for *D. melanogaster* Piwi, FBpp0079754 for *D. melanogaster* Aubergine, XP_039962265.1 for *B. tryoni* Piwi, XP_039954817.1 for *B. tryoni* Aubergine, XP_035903683.1 for *An. stephensi* Aubergine-Like, XP_035908796.1 for *An. stephensi* Aubergine, XP_001657626.2, XP_001653082.1, XP_001663870.2, XP_001652831.1, XP_021707306.1, and XP_001663409.1 for *Ae. aegypti* Piwi1, Piwi2, Piwi3, Piwi4, Piwi5, and Piwi6, respectively, BAF73718.2 for *Bombyx mori* Siwi, and XP_026300661.1 for *Apis mellifera* Aubergine-like, XP_008196304.2 for *Tribolium castaneum* Piwi-like, XP_039277686.1 for *Nilaparvata lugens* Piwi-like, XP_046398366.1 for *Ischnura elegans* Piwi-like, XP_076268964.1 and XP_076274955.1 for *Rhynchophorus ferrugineus* Piwi1 and Piwi2, respectively, XP_036139717.1, XP_012529020.2 and XP_036145939.1 for *Monomorium pharaonis* Piwi1, Piwi2 and Piwi3, respectively, XP_035434685.2, XP_050562113.1 and XP_050561924.1 for *Spodoptera frugiperda* Piwi1, Piwi2 and Piwi3, respectively. AGO3 from *D. melanogaster* (FBpp0289159), *B. mori* (BAF98575.1) and *T. castaneum* (XP_064215158.1), and AGO1 from *D. melanogaster* (FBpp0086739), *B. mori* (NP_001095931.1) and *T. castaneum* (XP_015837987.2) were also included. Aubergine/Piwi homologs in the *E. tenax*, *T. carbonaria*, *A. domesticus* and *T. oceanicus* genomes were predicted by tBLASTn using the homologs from *D. melanogaster*, *A. mellifera*, and *S. gregaria* as baits. Protein sequences were reconstructed from the following genomic regions: HG993125.1:41067699-41079689(-) for *E. tenax* Piwi, HG993125.1:55749476-55756254(-) for *E. tenax* Aubergine,

R2_4:13362229-13369035(-) for *T. carbonaria* Aubergine-Like, JAHLJT010009355.1:16831000-16838000(+), JAHLJT010008446.1:225566000-225573000(-), JAHLJT010009355.1:104356765-104369270(+) for *A. domesticus* Piwi1, Piwi2 and Piwi3, respectively, CAXIVR010000006.1:19291727-19300009(+) and CAXIVR010000001.1:83661957-83670227(+) for *T. oceanicus* Piwi1 and Piwi2, respectively, where chromosome or contig names with nucleotide positions and gene orientations are shown. N-PAZ-MID-PIWI domains were predicted by hhpred (https://toolkit.tuebingen.mpg.de/tools/hhpred), and multiple sequence alignment was performed using mafft-7.505 with the option '--auto'. The phylogenetic relationships were then inferred using iqtree-1.6.12 with the option '-m rtREV+R4 -bb 1000'.

## Data availability

Sequencing data and processed files have been deposited to Gene Expression Omnibus (GSE305623). Codes for the computational analyses, baits for the tBLASTn search of *gypsy* Open Reading Frames, reference genomic sequences of mosquito RNA viruses, multiple sequence alignments and the treefile of the phylogenetic analysis of PIWI proteins are made available at https://github.com/RippeiHayashi/insect_gypsy.

The source data of this paper are collected in the following database record: biostudies:S-SCDT-10_1038-S44319-026-00741-4.

## Peer review information

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

## Acknowledgements

We thank Peter Andersen (Aarhus University) for critical comments on the manuscript. We acknowledge Phillip Taylor from Macquarie University for sharing the Queensland fruit fly pupae. The following people contributed to the sample collection of other insects that were not included in the manuscript: Maciej Maselko (Macquarie University), Simon Baxter (University of Melbourne), Kanata and Keita Hayashi (Majura Primary School, Australian Capital Territory), and Azusa Hayashi (formerly affiliated to the Australian National University). This work was supported by the Australian Research Council (DP210102385).

## Author contributions

**Shashank Chary**: Resources; Formal analysis; Investigation; Writing—review and editing. **Patricia E Carreira**: Resources. **Sarah Nicholas**: Resources. **Kathryn B McNamara**: Resources. **Ian A Cockburn**: Resources. **Karin Nordström**: Resources. **Therésa M Jones**: Resources. **Rosalyn Gloag**: Resources. **Alyson Ashe**: Resources. **Leon E Hugo**: Resources. **Rippei Hayashi**: Conceptualization; Resources; Formal analysis; Supervision; Funding acquisition; Investigation; Writing—original draft; Writing—review and editing.

Source data underlying figure panels in this paper may have individual authorship assigned. Where available, figure panel/source data authorship is listed in the following database record: biostudies:S-SCDT-10_1038-S44319-026-00741-4.

## Disclosure and competing interests statement

The authors declare no competing interests.

# Expanded View Figures

**Figure EV1. Putative somatic piRNA clusters of *Eristalis tenax* and *Bactrocera tryoni*.**

(A) A scatter plot of the second replicate of ovarian and embryonic small RNA libraries of *E. tenax* showing the abundance of piRNAs from the ovaries (X axis) and the embryos (Y axis) that uniquely mapped to the individual 0.5 kb tiles. Tiles from newly identified clusters are highlighted in different colours. (B, C) piRNA coverage plots across the two other putative somatic piRNA clusters of *E. tenax*—HG993127.1 left in (B) and HG993129.1 in (C)—shown in counts per million genome mappers (cpm) from replicate 1 of the ovarian small RNA libraries. Reads that mapped to the plus- and minus-strands are coloured in dark and light grey, respectively. (D, D') Scatter plots of the first and second replicates of ovarian and embryonic small RNA libraries of *B. tryoni* comparing the abundance of piRNAs from the ovaries (X axis) and the embryos (Y axis) that uniquely mapped to the individual 0.5 kb tiles. Tiles from newly identified clusters are highlighted in different colours. (E, F) piRNA coverage plots across the two putative somatic piRNA clusters of *B. tryoni*—NC_052502.1 in (B) and NC_052503.1 in (C)—shown in counts per million genome mappers (cpm) from replicate 1 of the ovarian small RNA libraries. Reads that mapped to the plus- and minus-strands are coloured in dark and light grey, respectively. For (B), (C), (E) and (F), coloured bars indicate *gypsy* insertions predicted by RepeatMasker and *gypsy* GAG, POL and ENV open reading frames predicted by tBLASTn.

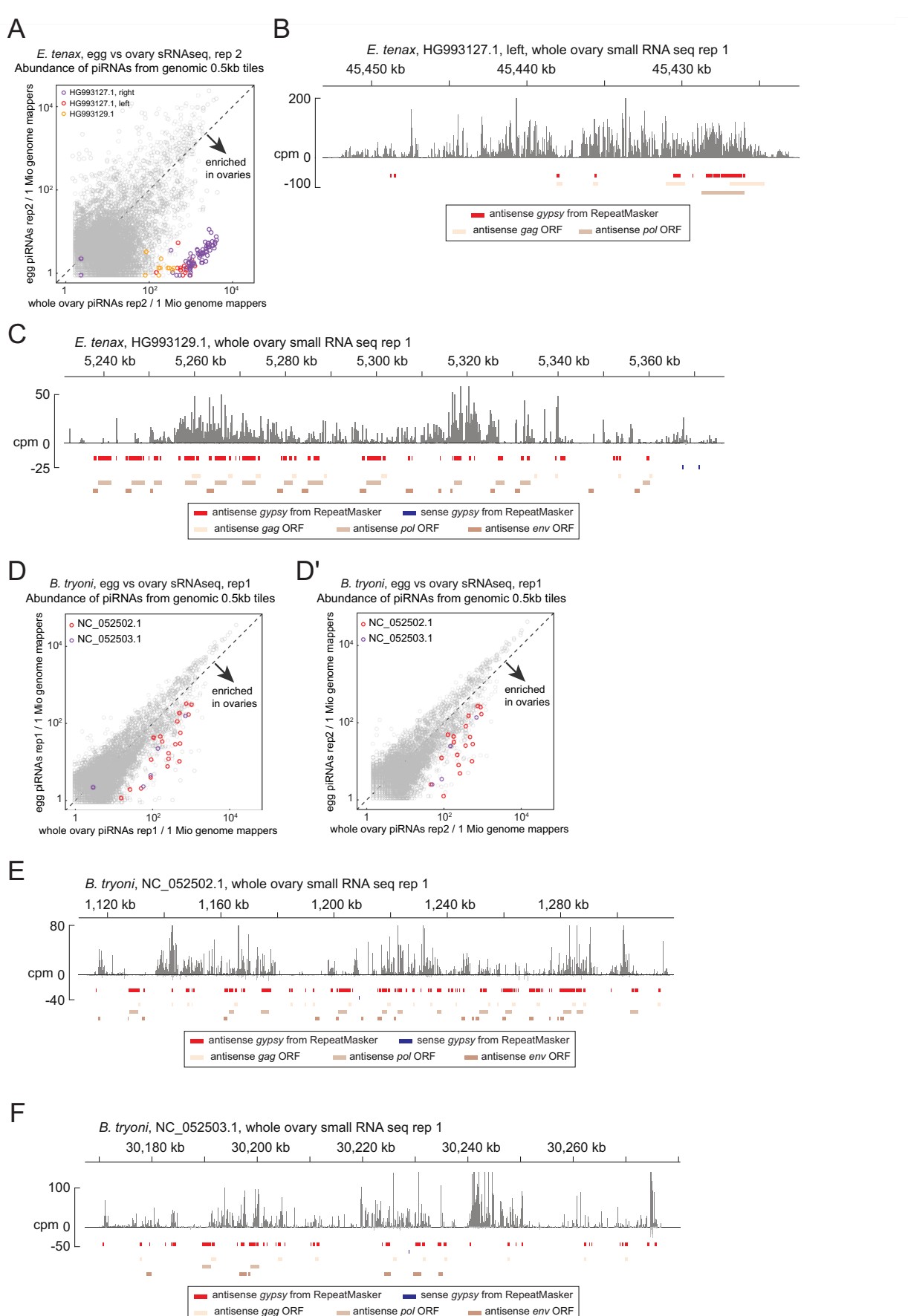

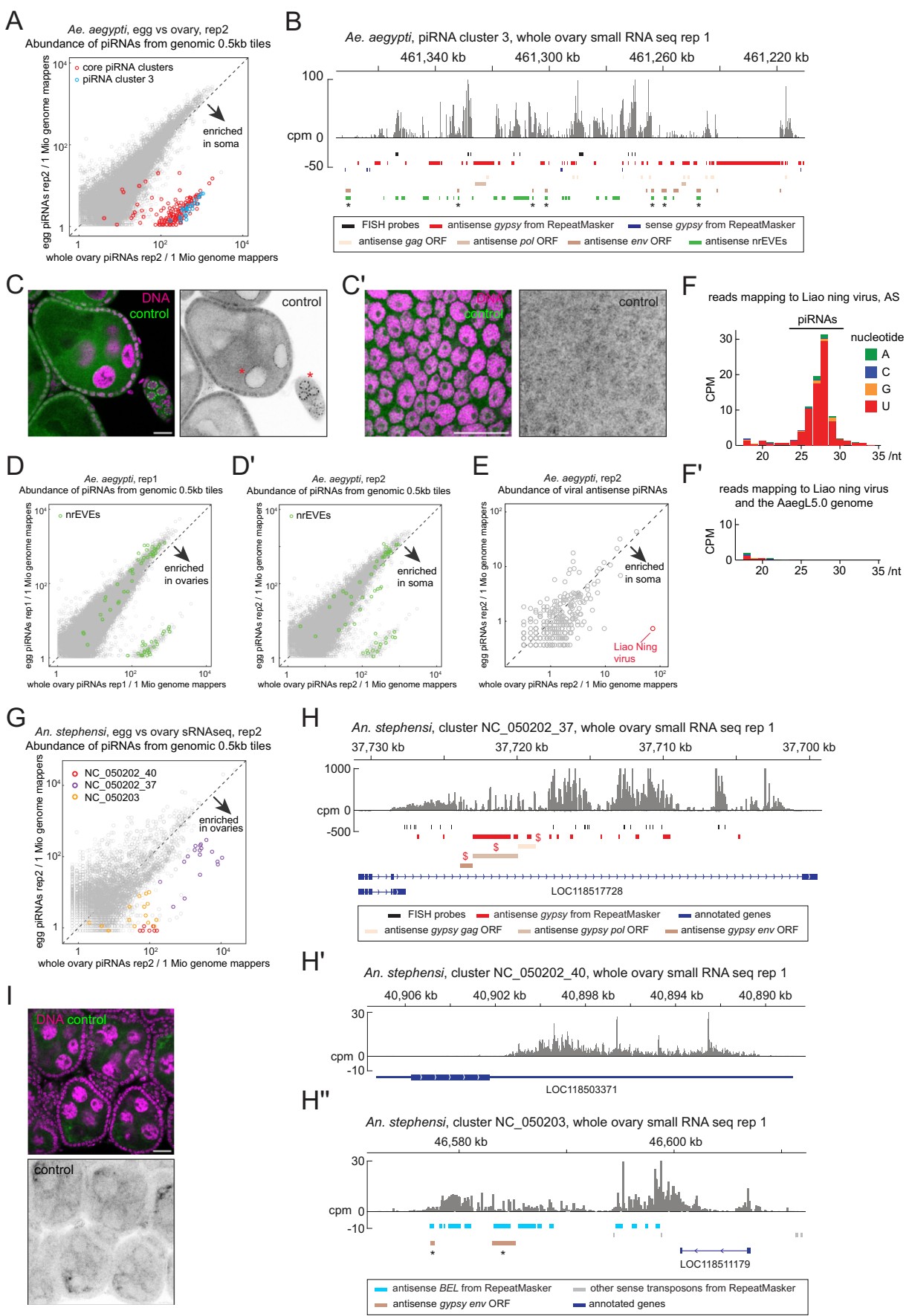

**Figure EV2.  Characterisation of the ovarian somatic piRNA clusters of *Aedes aegypti* and *Anopheles stephensi*.**

(A) Scatter plots for the replicate 2 of the small RNA sequencing of *Ae. aegypti* showing the ovarian somatic piRNA cluster tiles, equivalent to Fig. 2A. (B) Shown is the coverage of piRNA reads (> 22nt) in counts per million genome mappers (cpm) from the ovarian small RNA library of *Ae. aegypti* that uniquely mapped to the piRNA cluster 3. Reads that mapped to the plus- and minus-strands are coloured in dark and light grey, respectively. Coloured bars indicate *gypsy* insertions predicted by RepeatMasker, *gypsy* GAG, POL and ENV open reading frames predicted by tBLASTn, non-retroviral endogenous viral elements (nrEVEs) from Russo et al (2019) and the FISH probes. Some regions are predicted for *gypsy* elements and RNA viruses because they both have type III fusion glycoproteins (indicated by asterisks). (C, C') Side (C) and bird's-eye (C') views of RNA FISH images of *Ae. aegypti* egg chambers using probes against *A. domesticus piwi1* mRNA as a negative control. No specific signals are observed except for the staining around the germline nuclei (asterisks). Scale bars = 20 µm. (D, D') Scatter plots as in (A and Fig. 2A) but highlighting tiles in which more than half of the regions are occupied by nrEVEs. (E) Scatter plot for the replicate 2 showing the abundance of piRNAs mapping to viruses, equivalent to Fig. 2D. (F, F') Shown are the size distribution and 5'-end nucleotide frequency of small RNA reads mapping to the antisense strand of Liao ning virus segment 5 in (F), and of those additionally mapping to the *Ae. aegypti* genome in (F'). (G) A scatter plot for the replicate 2 of the small RNA sequencing of *An. stephensi*, equivalent to Fig. 2G, showing the ovary-enriched somatic piRNA cluster tiles. (H–H") piRNA coverage plots across three putative somatic piRNA clusters of *An. stephensi*—NC_050202_37 in (H), NC_050202_40 in (H'), and NC_050203 in (H")—shown in counts per million genome mappers (cpm) from replicate 1 of the ovarian small RNA library. Reads that mapped to the plus- and minus-strands are coloured in dark and light grey, respectively. Coloured bars indicate *gypsy* and *BEL* insertions predicted by RepeatMasker and *gypsy* GAG, POL and ENV open reading frames predicted by tBLASTn, annotated genes and the FISH probes. Some regions are predicted for *gypsy* and *BEL* because they both have type III fusion glycoproteins (indicated by asterisks). Antisense *gypsy* fragments that were examined for corresponding full-length insertions in Fig. EV5 are marked by dollars. (I) A control RNA FISH image of *An. stephensi* egg chambers using probes against *A. domesticus piwi1* mRNA. Scale bar = 20 µm. Source data are available online for this figure.

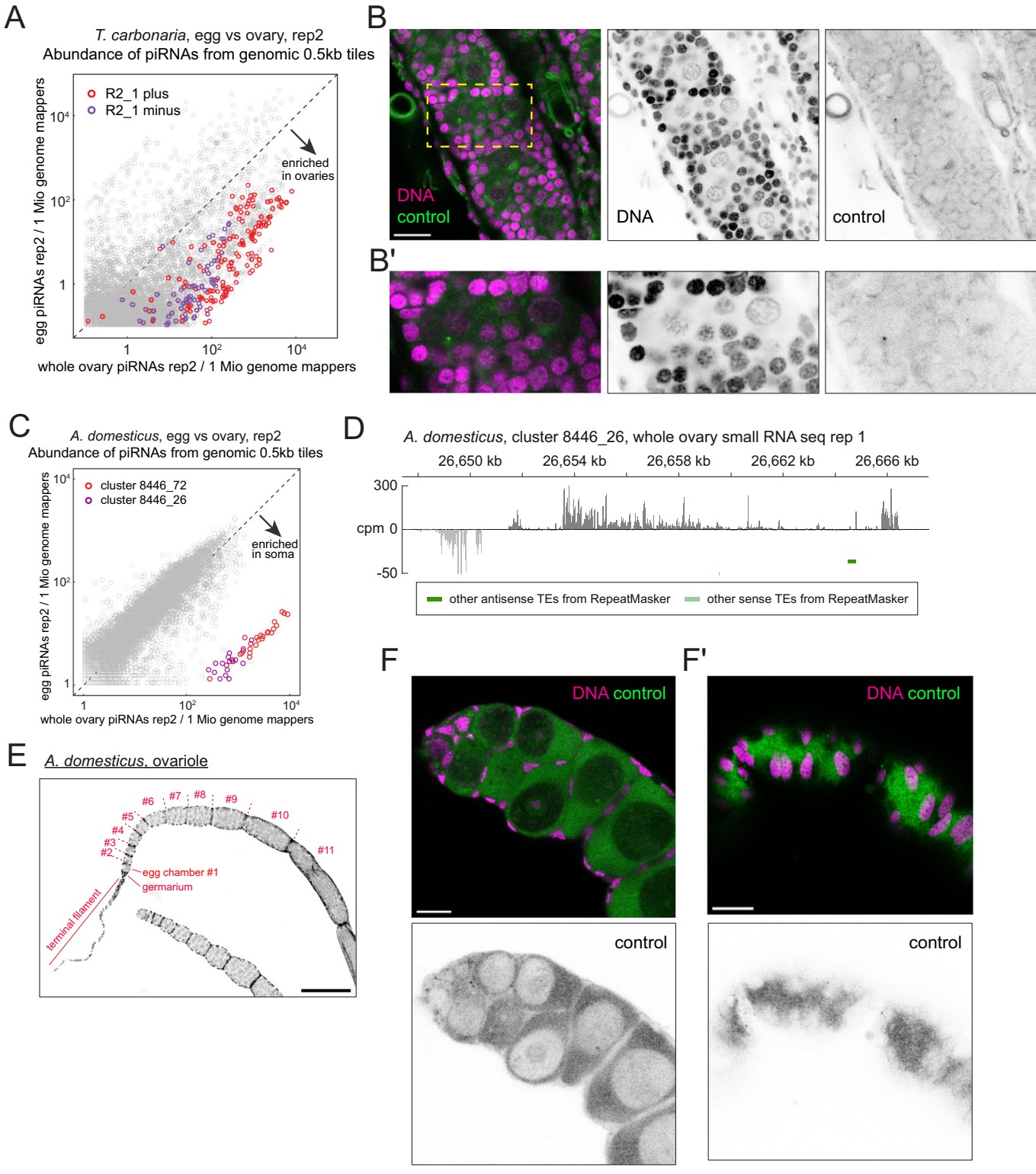

Figure EV3.   Characterisation of the ovarian somatic piRNA clusters of stingless bees and crickets.

(A) A scatter plot for the replicate 2 of the small RNA sequencing of *T. carbonaria*, equivalent to Fig. 3A, showing the tiles from the ovary-enriched somatic piRNA cluster R2_1. (B) A control RNA FISH image of *T. carbonaria* egg chambers using probes against *A. domesticus piwi1* mRNA. (C) A scatter plot for the replicate 2 of the small RNA sequencing of *A. domesticus*, equivalent to Fig. 4A, showing the ovary-enriched somatic piRNA cluster tiles. (D) A piRNA coverage plot across the somatic piRNA cluster 8446_26 of *A. domesticus* shown in counts per million genome mappers (cpm) from replicate 1 of the ovarian small RNA library. Reads that mapped to the plus- and minus-strands are coloured in dark and light grey, respectively. (E) A confocal image showing an ovariole of *A. domesticus* stained by DAPI. The terminal filament, germarium and developing egg chambers are indicated. (F, F') Control RNA FISH images of *A. domesticus* egg chambers using probes against *D. melanogaster piwi* mRNA. Scale bars = 20 μm. Source data are available online for this figure.

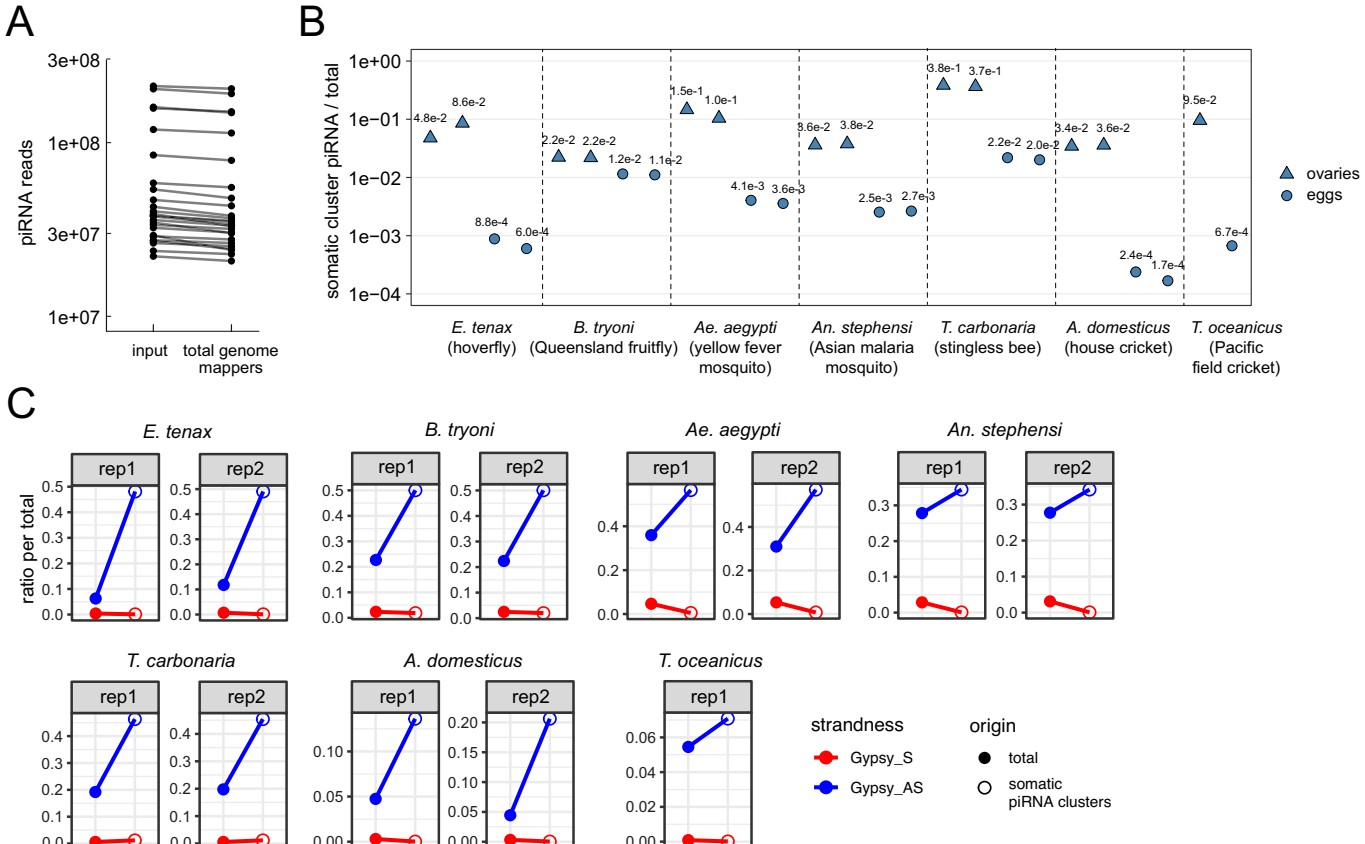

**Figure EV4. Ovarian somatic cells selectively express antisense *gypsy* piRNAs across insects.**

(A) Shown are the number of reads (> 22nt) sequenced (input) and mapped to the respective genomes of all 26 small RNA sequencing libraries analysed in this study. (B) Shown are the fraction of piRNA reads (> 22nt) from ovarian and embryonic small RNA sequencing mapping to the ovarian somatic piRNA clusters of the respective genomes out of all genome mapping piRNAs. (C) Shown are the fraction of piRNA reads originating from sense and antisense strands of *gypsy* insertions out of all genome-mapping piRNAs. The coverage of reads mapping to multiple genomic loci was normalised by the number of mapping instances per read for (B) and (C).

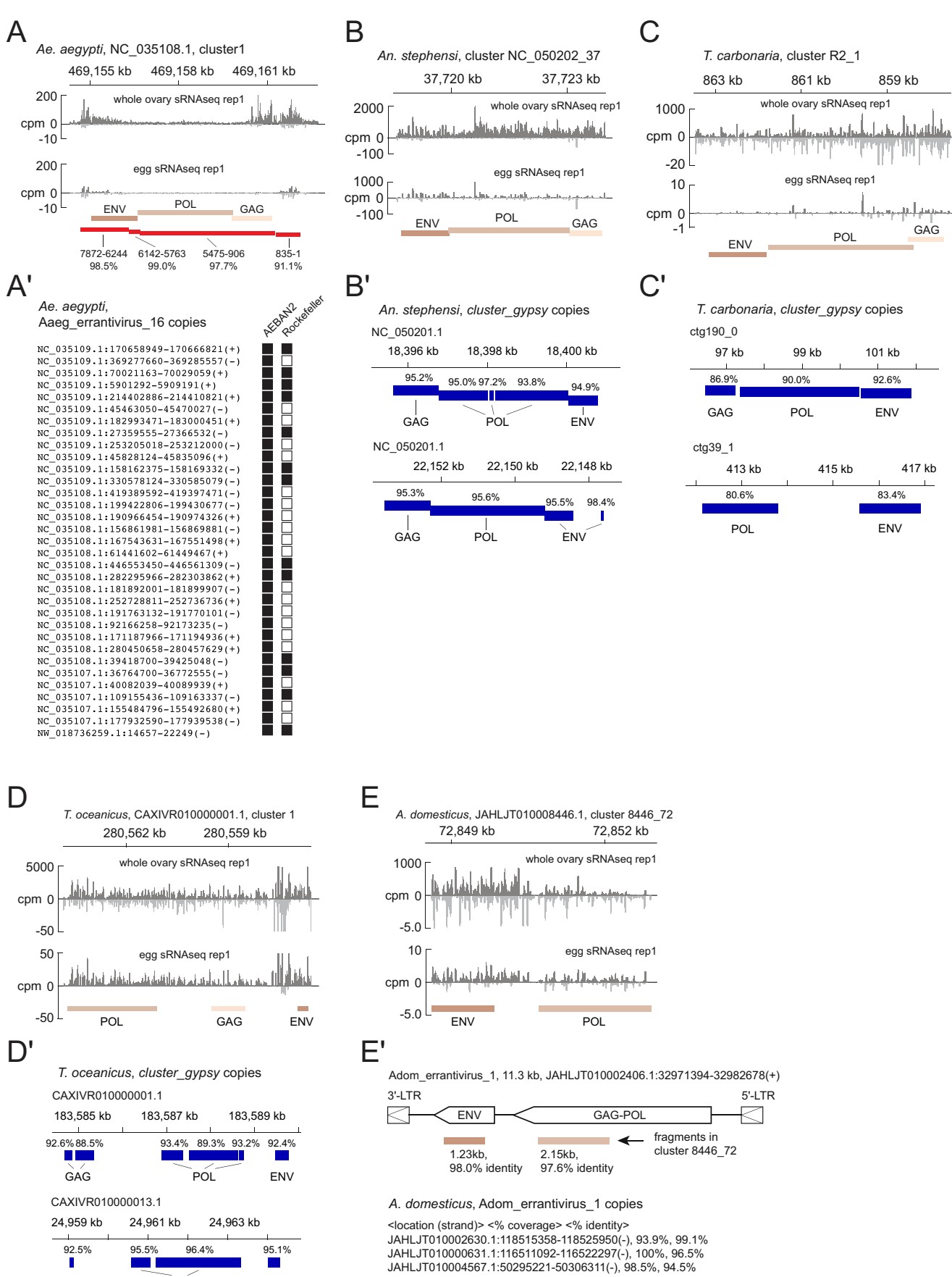

**Figure EV5.  Genomic *gypsy* insertions that are targeted by ovarian somatic piRNAs.**

(A) Shown are *gypsy* GAG, POL and ENV insertions in cluster 1 of the *Ae. aegypti* genome that are homologous to the previously identified *env*-containing *gypsy* element *Aaeg_errantivirus_16* (Chary and Hayashi, 2025). Nucleotide positions and percent sequence identities of the homologous regions are shown. The piRNA read coverage from ovarian and embryonic small RNA libraries is shown. The coverage of reads mapping to multiple genomic loci was normalised by the number of mapping instances per read across all panels in this figure. (A') Shown are the genomic positions of 33 *Aaeg_errantivirus_16* copies in the AaegL5.0 genome assembly (Liverpool strain). Presence and absence of the syntenic copies in the AEBAN2 isolate (IBAB_Aaeg_KPA_1.0) and the Rockefeller strain (CU_AaegROCK_1.0) are shown in filled and open boxes, respectively. (B–E) Shown are the piRNA read coverage of the ovarian and embryonic small RNA sequencing libraries from *An. stephensi* (B), *T. carbonaria* (C), *T. oceanicus* (D) and *A. domesticus* (E), across the *gypsy* GAG, POL and ENV insertions that are tandemly aligned in ovarian somatic piRNA clusters. Positions of these insertions in the cluster are shown in Fig. EV2 for *An. stephensi*, Fig. 3 for *T. carbonaria* and Fig. 4 for *T. oceanicus* and *A. domesticus*. (B'–E') Shown are *gypsy* copies of the respective genomes harbouring GAG, POL and ENV open reading frames that are each homologous to the insertions found in the piRNA clusters. Sequence identifies to the cluster insertions are shown in percentage. Cluster insertions of ENV and POL in *A. domesticus* are homologous to an *env*-containing *gypsy* element, which we named as *Adom_errantivirus_1*. The *A. domesticus* genome carries three copies of *Adom_errantivirus_1* with >90% coverage and >90% sequence identities.

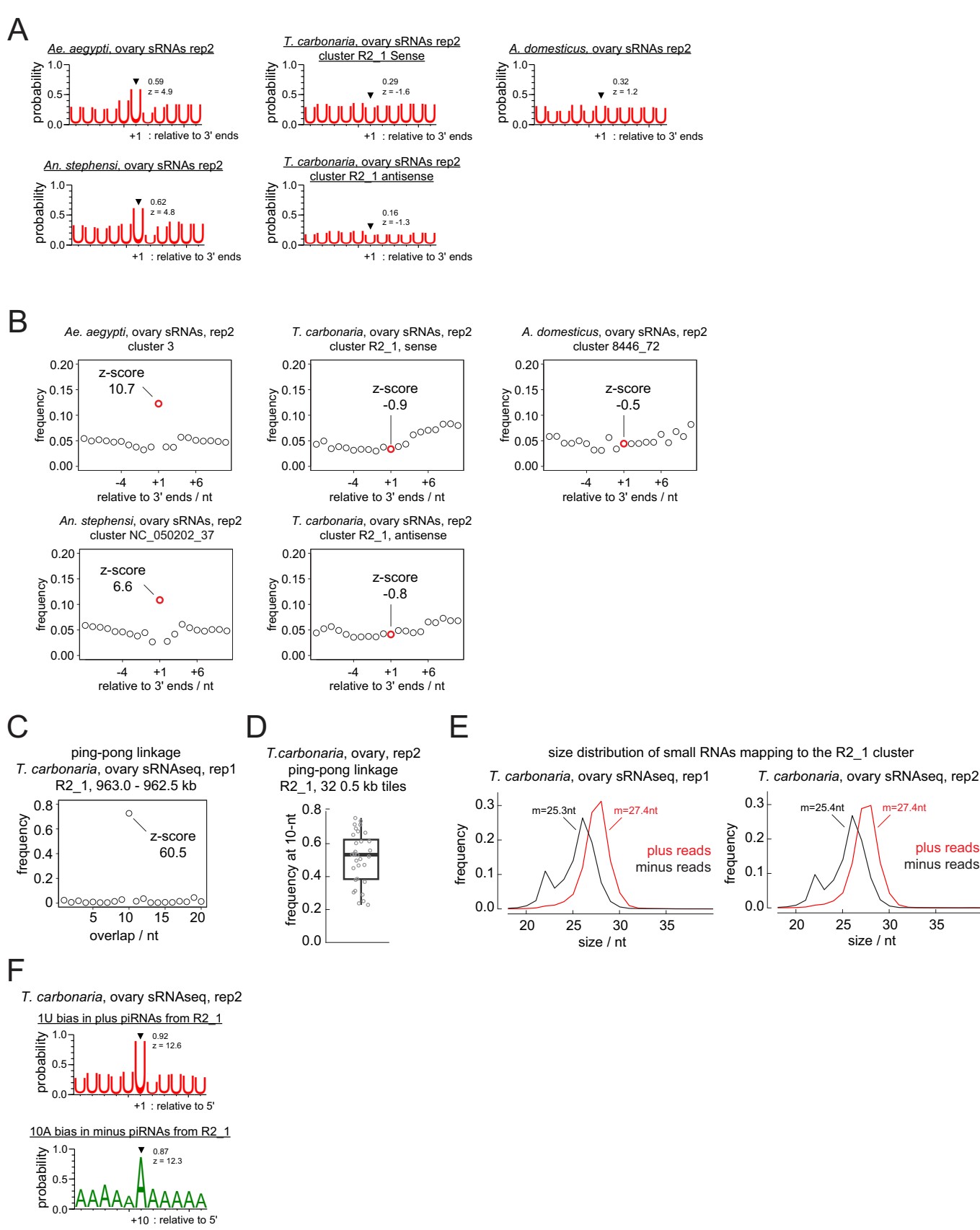

Figure EV6. **Linkage analysis of phasing and ping-pong piRNA biogenesis for ovarian small RNA libraries.**

(A, B) Analyses performed in Fig. 6 were repeated for the second replicates of ovarian small RNA libraries from different insects. Shown in (A) are the frequencies of uridines found at positions relative to the 3′ ends of piRNAs mapping to the ovarian somatic piRNA clusters of individual insect species. Frequencies at the linkage position (+1) and the z-scores are shown. In (B), frequency plots of the distance between piRNA 3′ and 5′ ends from representative ovarian somatic clusters are shown for individual species. The Z-scores of the linkage distance +1, in which piRNA 5′ ends are found immediately after piRNA 3′ ends, are shown. (C) Frequency plot of the 5′–5′ overlaps of piRNAs from the plus- and minus-strands of *T. carbonaria* cluster R2_1, region 963.0 - 962.5 kb. The Z-score of the linkage overlap length—10 nucleotides—is shown. (D) Boxplot showing frequencies at the 10-nt overlap from 0.5 kb tiles from cluster R2_1. Circles represent values from individual 0.5 kb genomic tiles from the clusters. The centre line indicates the medians; box limits represent the interquartile range (IQR); whiskers extend to 1.5× IQR. $N = 32$. (E) Size distributions of the two replicates of *T. carbonaria* ovarian small RNAs mapping to the plus- and minus-strands of cluster R2_1 are shown with mean lengths. (F) Shown are the frequencies of Uridines and Adenosines found at positions around the 5′ end and the tenth nucleotide of piRNAs mapping to the plus- and minus-strands of cluster R2_1. Frequencies at the linkage positions and the z-scores are shown. The analyses performed in Fig. 7C,E were repeated for the second replicate of *T. carbonaria* ovarian small RNA library in (D) and (F), respectively.

## A

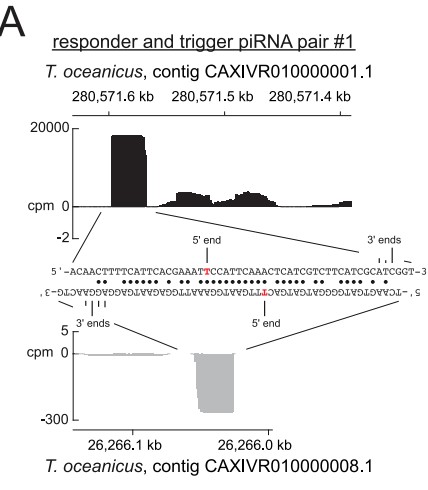

responder and trigger piRNA pair #1

*T. oceanicus*, contig CAXIVR010000001.1

*T. oceanicus*, contig CAXIVR010000008.1

## B

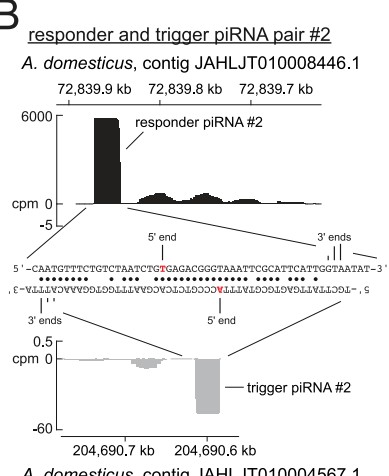

responder and trigger piRNA pair #2

*A. domesticus*, contig JAHLJT010008446.1

*A. domesticus*, contig JAHLJT010004567.1

## C

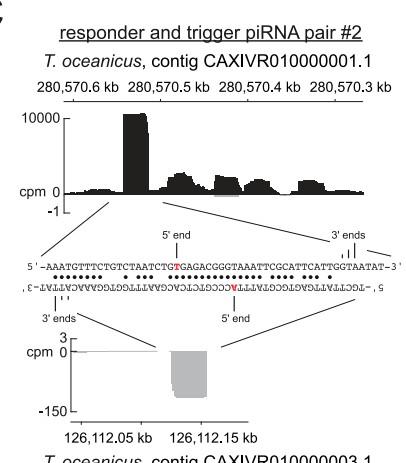

responder and trigger piRNA pair #2

*T. oceanicus*, contig CAXIVR010000001.1

*T. oceanicus*, contig CAXIVR010000003.1

## D

conservation of genomic sequences around responder and trigger piRNAs

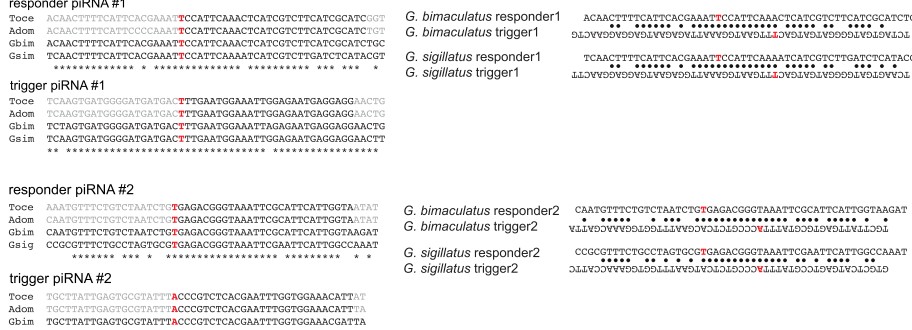

## D'

distance between responder piRNAs #1 and #2:

*T. oceanicus*, CAXIVR010000001.1, 1055 nt
*A. domesticus*, JAHLJT010008446.1, 1898 nt
*G. bimaculatus*, OZ281568.1, 1874 nt
*G. sigillatus*, CBCOPG010000001.1, 12603 nt

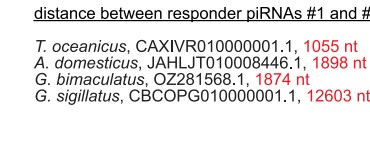

## D''

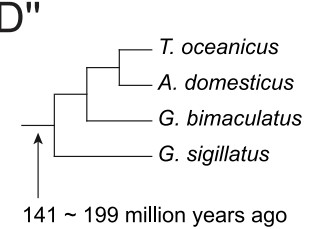

141 ~ 199 million years ago

## E

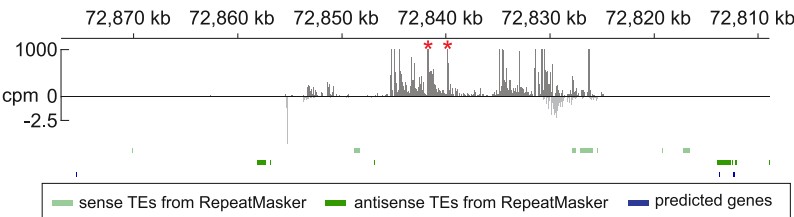

*A. domesticus*, cluster 8446_72 with predicted gene annotations

## F

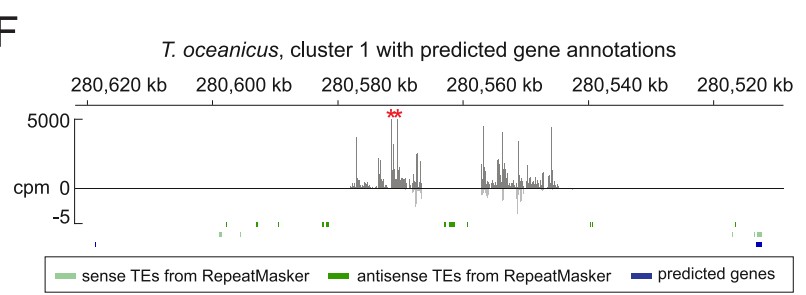

*T. oceanicus*, cluster 1 with predicted gene annotations

**Figure EV7.  Conversation of trigger and responder piRNA pairs in cricket ovarian somatic clusters.**

(A–C) Sequences and the cpm coverage of genomic regions at responder and trigger piRNA pairs #1 from *T. oceanicus* in (A), pairs #2 from *A. domesticus* in (B) and *T. oceanicus* in (C). The 5′ and 3′ end positions are indicated with frequencies represented by heights for the 3′ ends. Watson-Crick base pairs are marked by circles. (D) Sequence alignments of the genomic regions at the responder and trigger piRNAs from various cricket species. The two responder piRNAs are derived from the same genomic locus for all four species. (D′) Distances between them for each species are shown. (E, F) The flanking genomic regions of the piRNA clusters possessing the two responder piRNAs (asterisks) are shown for *A. domesticus* in (E) and *T. oceanicus* in (F) with protein-coding genes predicted by BLASTn search of previously published transcriptome of each species (Bailey et al, 2013; Oppert et al, 2020).

