## [Peer Review File · EMBO Reports]

Conserved but mechanistically diverse piRNA defence against endogenous retroviruses in insects

Rippe Hayashi, Shashank Chary, Patricia Carreira, Sarah Nicholas, Kathryn McNamara, Ian Cockburn, Karin Nordström, Thérésa Jones, Rosalyn Gloag, Alyson Ashe, and Leon Hugo

Corresponding author(s): Rippei Hayashi (rippei.hayashi@anu.edu.au)

Review Timeline:

Submission Date:	8th Sep 25
Editorial Decision:	27th Oct 25
Revision Received:	5th Jan 26
Editorial Decision:	12th Feb 26
Revision Received:	13th Feb 26
Accepted:	23rd Feb 26

Editor: Yehu Moran

Transaction Report:

Dear Dr. Hayashi

Thank you for the submission of your manuscript to EMBO reports. We have now received the full set of referee reports as well as referee cross-comments that are all pasted below.

As you will see, the referees acknowledge that the findings are interesting. However, they also raise concerns that require your attention.

I would thus like to invite you to revise your manuscript with the understanding that the referee concerns must be fully addressed and their suggestions taken on board, whenever possible. Please address all referee concerns in a complete point-by-point response. Acceptance of the manuscript will depend on a positive outcome of a second round of review. It is EMBO reports policy to allow a single round of major revision only and acceptance or rejection of the manuscript will therefore depend on the completeness of your responses included in the next, final version of the manuscript.

We realize that it is difficult to revise to a specific deadline. In the interest of protecting the conceptual advance provided by the work, we recommend a revision within 3 months (27th Jan 2026). Please discuss the revision progress ahead of this time with the editor if you require more time to complete the revisions.

- 1) A data availability section providing access to data deposited in public databases is missing. If you have not deposited any data, please add a sentence to the data availability section that explains that.
- 2) Your manuscript contains statistics and error bars based on $n=2$. Please use scatter blots in these cases. No statistics should be calculated if $n=2$.

<<https://www.embopress.org/page/journal/14693178/authorguide#expandedview>>

5) a complete author checklist, which you can download from our author guidelines

<<https://www.embopress.org/page/journal/14693178/authorguide>>. Please insert information in the checklist that is also reflected in the manuscript. The completed author checklist will also be part of the RPF.

6) Please note that all corresponding authors are required to supply an ORCID ID for their name upon submission of a revised manuscript (<<https://orcid.org/>>). Please find instructions on how to link your ORCID ID to your account in our manuscript tracking system in our Author guidelines <<https://www.embopress.org/page/journal/14693178/authorguide#authorshipguidelines>>

7) Before submitting your revision, primary datasets produced in this study need to be deposited in an appropriate public database (see <https://www.embopress.org/page/journal/14693178/authorguide#datadeposition>). Please remember to provide a reviewer password if the datasets are not yet public. The accession numbers and database should be listed in a formal "Data Availability" section placed after Materials & Method (see also <https://www.embopress.org/page/journal/14693178/authorguide#datadeposition>). Please note that the Data Availability Section is restricted to new primary data that are part of this study. * Note - All links should resolve to a page where the data can be accessed. *
If your study has not produced novel datasets, please mention this fact in the Data Availability Section.

12) All Materials and Methods need to be described in the main text using our 'Structured Methods' format, which is required for all research articles. According to this format, the Methods section includes a Reagents and Tools Table (listing key reagents, experimental models, software and relevant equipment and including their sources and relevant identifiers) followed by a Methods and Protocols section describing the methods using a step-by-step protocol format. The aim is to facilitate adoption of the methodologies across labs. More information on how to adhere to this format as well as a downloadable template (.docx) for the Reagents and Tools Table can be found in our author guidelines: <https://www.embopress.org/page/journal/14693178/authorguide#structuredmethods>.

An example of a Method paper with Structured Methods can be found here: <https://www.embopress.org/doi/full/10.1038/s44320-024-00037-6#sec-4>

I look forward to seeing a revised form of your manuscript when it is ready.

Yours sincerely,

Yehu Moran
Editor
EMBO Reports

Referee #1:

The manuscript by Chary et al investigates the flux of piRNAs-mediated defense against gypsy long terminal repeat (LTR) retrotransposons that propagate from somatic elements of the gonad. These TEs are competent to form infectious particles, which enables this life cycle. Originally observed in *Drosophila* follicle cells, a unique configuration of the piRNA pathway represses this class in *Drosophila* to prevent infection of Oocytes. Given the widespread presence of these TEs, the authors sought if a similar mechanism is at play in other Dipterans, as well as Hymenopterans, and Orthopterans (flies, bees, and crickets). The main approach was to compare the expression of piRNAs in oocytes versus ovaries using the assumption that only libraries generated from ovaries would contain RNAs from somatic gonad. From this, the authors do find that gypsy-style TE-targeting piRNAs are present, showing that the mode of invasion in question is an ancient strategy that is counteracted by piRNAs. In the process of analyzing datasets from different species the authors note that different modes of piRNA biogenesis are apparently used in different clades to suppress the gypsy-like TEs.

Overall, the work establishes a common mode of transposon action that insects suppress through piRNAs. This is interesting given the loss of piRNA expression outside of gonads in some insects (ie *D melanogaster*) and highlights the pressure on genome surveillance pathways to continuously suppress gypsy-like TEs in somatic gonad cells. Data presented figures 1-5 and its interpretation is for the most part high quality. However, I would recommend the authors find a way to condense these results for a streamlined narrative. See comments below. A bigger issue is the presentation of data in figure 6 which has 16 panels of data where different analyses are performed for different species. This compromises the most important conclusions of the article concerning different modes of piRNAs being used in different species. The authors should consistently apply analyses like ping pong and phasing assessment to all datasets. Also, if the authors want to make sweeping statements about piRNA function in insects I recommend they include in their study all available relevant public datasets in their analyses. Expansion of the study to all available species will enhance interest in the article. Therefore, while I think the article is appropriate for publication in EMBO due to the wide-ranging implications I recommend a major revision to enhance the conclusions.

Major concerns:

- The primary issue is the disjointed approach to data analysis in the article presented in figure 6. The most important conclusions of the article are shown in figure 6J, but unfortunately, key analyses such as pingpong and phasing analysis shown in Figures 6D', C, E, and G are not performed for loci of interests for data sets from all species. They are only shown are *T. carbonaria*, and mosquitos. The authors should run these analyses for a side-by-side comparison for all species to support the conclusions in part J.
- The authors make the claim that slicing-independent phasing is occurring in mosquitos by showing the expression of a Aub-like homolog in gonadal soma. This does not definitely show the mode of phasing is the same as *drosophila* where *dme Piwi* is the sole PIWI protein in follicle cells which lacks slicer. The authors need to show there are no responder piRNAs involved to make this claim about mosquitos.
- The analysis of Liao Ning virus needs further support. The authors use a genome assembly from a naïve strain of *A. aegypti*. This leaves the possibility there has been insertion of viral sequences in the strain where small RNAs were sequenced. The authors need to show that this is not a case to make claims about the origin of the lao ning-derived piRNAs.
- The authors should mine public databases for all similar datasets to expand the reach of the conclusions of the article.
- The phylogeny shown in Fig 6A would benefit from adding more orthologs from species across insecta. This would better illustrate the relationships of the PIWI proteins of interest. The phylogeny should also include Ago1 orthologs to reinforce the identity of the genes of interest are in fact PIWIs.

Minor comments

- No description of phasing analysis in methods
- The authors may consider consolidating figures and moving genome density plots to supplement to focus readers on most important data points.

- The authors should include scientific names and common names of the species analyzed in the abstract to help readers be clear as to which species is which
- The authors should add common names to figures to help reader understand the significance of data points

Referee #2:

The authors present a very interesting comparative analysis of ovary somatic cell piRNA expression across multiple insect species. By profiling small RNAs from the ovaries and eggs of mosquitoes, bees, crickets, and *Drosophila*, and then focusing on just the ovary-enriched piRNAs that could represent the somatic compartment, the authors argue that somatic piRNA defenses against gypsy retrotransposons have been maintained for over 400 million years, while diverging in their biogenesis mechanisms. This work provides valuable evolutionary insight into small RNA-based genome defense and its potential relationship to antiviral responses. The study is novel though it would benefit from some revisions to the issues below to strengthen mechanistic evidence and enhance quantitative analysis.

1. Authors description of somatic piRNA abundance from the enrichment method needs additional quantification and description of cutoffs to indicate how deep is the capture of somatic piRNAs since this analysis method relies on subtracting egg piRNAs from ovary piRNAs to use enrichment to represent somatic follicle cell-contributing piRNAs. This enrichment is easy to observe in the scatterplots of all the other insects except for the bee *T. carbonaria*, where piRNAs enriched in the ovaries is more murky and indistinct compared to the other insects where all the ovary enriched piRNAs are quite distinct. This issue is relevant to the next point because if the ovary-specific enrichment in bee ovaries is not clear, it could be likely a lot more germline piRNAs greatly influenced by ping-pong is dominating the signal of ping pong in Fig. 6 for bees.
2. The assessment in Fig 6 that the likely somatic compartment of the bee is just ping pong and not at all any phasing is not completely convincing by what is presented in Fig. 6, since there is just the Fig 6 D-F, but where is the negative evidence of no phasing at all? We do not see a phasing analysis diagram like what has been shown in Fig. 2 of Gainetdinov et al PMID: 30193099 or Fig. 7 of Ma et al PMID: 33419731 that would rule out phasing. It would also be helpful to readers to include in parentheses by the insect species name the colloquial name for at least one panel when there are many different species being compared. For example in Fig. 6, *A. domesticus* (house cricket), *T. carbonaria* (stingless bee), *T. oceanicus* (field cricket). If the authors cannot convincingly strengthen the claim of no somatic piRNA phasing in bees, the conclusions in the discussion around somatic piRNA biogenesis in bees in the Discussion should also be tempered.
3. The authors need to provide better normalized quantitative of the reads per million mapping to gypsy versus other piRNA clusters as a fraction of all small RNAs being the enriched fraction of the ovary versus the egg when comparing pathway strength among species. This will strengthen the claim of conserved piRNA defense rather than simply shared presence.
4. The rendition of genomic strand piRNA mapping and Sense/Antisense orientation in the Figures of 1D and Fig. 2B and Fig. 3B, 3B' and 3B' and Fig. 4B and Fig. 5B, 5B' and 5C and Fig. S1, S2, could be improved since this is salient to the description of the other insect piRNA clusters being similar to the *Dmel* Flamenco piRNA cluster. The current state of the figure is to present "plus- and minus-strands are coloured in dark and light gray, respectively" which is challenging to discern. Authors should follow the convention of red and blue for plus and minus strand read mapping to allow readers to discern the polarity bias. Also, arrows for the gypsy transposon elements would help delineate if the TEs are mapping in plus or minus strand orientation. A much better arrangement of this figure would help the reader see if the conclusion is true that the other insect piRNA are primarily antisense to the gypsy elements.
5. The mapping of the *aedes aegypti* ovarian piRNAs to Liao Ning Virus (LNV) is interesting, but the Fig. 2E, 2F and 2G and unclear how significant are these mappings to indicate that their *Ae aegypti* strain is being truly infected by LNV or if instead their Australian strain of *Ae. aegypti* just has taken up segment 5 of LNV into its genome as a potential new Endogenous Viral Element (EVE). We recommend the authors see the study by Ma et al PMID: 33419731 where a strong insect virus infection in mosquitoes should generate viral piRNA through the plus and minus mapping reads of the virus genome, whereas this example is only to segment 5 but not any of the other 11 segments of the LNV genome? Also, the vast majority of these LNV viral piRNAs are just AntiSense, and the authors should examine and see also a strong bias for AS piRNAs to the nrEVEs. When the authors have properly compared nrEVE piRNAs to this putative LNV-targeting piRNAs, we recommend the authors revise their text to acknowledge they do not have direct evidence of an actual LNV infection in their *Ae. aegypti* strain but more likely they are observing an appearance of an EVE based off of an LNV segment.
6. There should be a small RNA library sequencing statistics and mapping statistics table in the supplement showing that ovary and egg libraries are all sequenced to the equivalent and sufficient depth. If some of the small RNA libraries are sequenced to be below 10M reads those should be noted as potentially too shallow.
7. Authors view of "shared origin of antiviral and anti-retrotransposon defenses in *Aedes aegypti*" is intriguing but not directly tested. This claim needs to be supported by additional evidence for overlap or shared expression of clusters that produce both viral- and gypsy-targeting piRNAs, otherwise authors need to tone down the claim to "possible shared origin."

8. To support the authors claim of "long-term arms race" hypothesis, they need to include evidence of current or recent TE activity, such as polymorphic insertions or TE expression profiles. Without demonstrating ongoing mobility, the evolutionary narrative remains speculative.

9. The paper says "The coordinates of the piRNA clusters used in this study can be found in the git repository." But this table of piRNA cluster coordinates should really be part of this paper, not just in the git repository.

Dear Yehu,

We thank reviewers for critically evaluating our manuscript and providing constructive suggestions. We enclosed here the revised manuscript for your consideration and the letter highlighting the changes made in the manuscript in response to reviewers' comments.

We included new analyses and rearranged previous figure panels. All changes are reflected and made visible in the revised manuscript. The github repository (https://github.com/Rippeihayashi/insect_gypsy) has also been updated accordingly.

Sincerely,

Rippeihayashi

Referee #1:

The manuscript by Chary et al investigates the flux of piRNAs-mediated defense against gypsy long terminal repeat (LTR) retrotransposons that propagate from somatic elements of the gonad. These TEs are competent to form infectious particles, which enables this life cycle. Originally observed in *Drosophila* follicle cells, a unique configuration of the piRNA pathway represses this class in *Drosophila* to prevent infection of Oocytes. Given the widespread presence of these TEs, the authors sought if a similar mechanism is at play in other Dipterans, as well as Hymenopterans, and Orthopterans (flies, bees, and crickets). The main approach was to compare the expression of piRNAs in oocytes versus ovaries using the assumption that only libraries generated from ovaries would contain RNAs from somatic gonad. From this, the authors do find that gypsy-style TE-targeting piRNAs are present, showing that the mode of invasion in question is an ancient strategy that is counteracted by piRNAs. In the process of analyzing datasets from different species the authors note that different modes of piRNA biogenesis are apparently used in different clades to suppress the gypsy-like TEs.

Overall, the work establishes a common mode of transposon action that insects suppress through piRNAs. This is interesting given the loss of piRNA expression outside of gonads in some insects (ie *D. melanogaster*) and highlights the pressure on genome surveillance pathways to continuously suppress gypsy-like TEs in somatic gonad cells. Data presented figures 1-5 and its interpretation is for the most part high quality. However, I would recommend the authors find a way to condense these results for a streamlined narrative. See comments below. A bigger issue is the presentation of data in figure 6 which has 16 panels of data where different analyses are performed for different species. This compromises the most important conclusions of the article concerning different modes of piRNAs being used in different species. The authors should consistently apply analyses like ping pong and

phasing assessment to all datasets. Also, if the authors want to make sweeping statements about piRNA function in insects I recommend they include in their study all available relevant public datasets in their analyses. Expansion of the study to all available species will enhance interest in the article. Therefore, while I think the article is appropriate for publication in EMBO due to the wide-ranging implications I recommend a major revision to enhance the conclusions.

Major concerns:

- The primary issue is the disjointed approach to data analysis in the article presented in figure 6. The most important conclusions of the article are shown in figure 6J, but unfortunately, key analyses such as pingpong and phasing analysis shown in Figures 6D', C, E, and G are not performed for loci of interests for data sets from all species. They are only shown are *T. carbonaria*, and mosquitos. The authors should run these analyses for a side-by-side comparison for all species to support the conclusions in part J.

We now expanded the analysis on each piRNA biogenesis mechanism, applied that to all species studied in the manuscript, and presented them in three separate figures (Figure 6 to 8). We excluded hoverflies and Queensland fruit flies from the analysis because the *Drosophila* Piwi is conserved in these species, and therefore, the mechanistic conservation was expected. Since the newly identified ovarian somatic piRNA clusters predominantly produce piRNAs from single strand in most species (see Figure 7D), we only analysed the ping-pong biogenesis for *Tetragonula carbonaria* (stingless bees).

- The authors make the claim that slicing-independent phasing is occurring in mosquitos by showing the expression of a Aub-like homolog in gonadal soma. This does not definitely show the mode of phasing is the same as drosophila where dme Piwi is the sole PIWI protein in follicle cells which lacks slicer. The authors need to show there are no responder piRNAs involved to make this claim about mosquitos.

We performed *in-trans* ping-pong analysis to identify and quantify trigger-responder piRNA pairs (Figure 8C). The analysis showed that piRNAs expressed in the ovarian somatic clusters or soma-enriched (>7-fold enrichment for *Anopheles stephensi* and >10-fold enrichment for the other species) genomic tiles do not find pairs more than the background. This was the case for both *Aedes aegypti* and *Anopheles stephensi*.

A small number of trigger-responder piRNA pairs were found in the soma-enriched genomic tiles of *Anopheles stephensi* (data that are not marked by the asterisks are for the tile-focused analysis). We believe this was because the separation of the germline and somatic tiles was not prominent for *Anopheles stephensi*.

In the first manuscript, we analysed somatic cluster-derived piRNAs for *Aedes aegypti*. In the revised manuscript, we analysed piRNAs from soma-enriched genomic tiles to be consistent with the other species. The linkage values between the two analyses did not change: see the panel below only for the reviewers (data marked by the asterisks are for the cluster-focused analysis).

In the course of this analysis, we identified a single ping-pong piRNA pair TTTCTATTAATATGTTGCAATCAACCG and TTAATAGAAATTACAGTGGTTGGTATCTT both mapping to a tile spanning positions 351,125,500-351,126,000 on chromosome NC_035109.1 of *Aedes aegypti*. Further analysis revealed that piRNAs beginning with “TTTCTATTA” and their corresponding ping-pong pairs beginning with “TTAATAGAAA”, which mapped to the genome at multiple sites, were highly expressed both in the whole ovaries and embryos. Because canonical ping-pong piRNA pairs can artificially inflate in-trans ping-pong linkage signals, we excluded this tile from the analysis. The new method section for the in trans ping-pong linkage analysis has been updated reflecting these changes.

- The analysis of Liao Ning virus needs further support. The authors use a genome assembly from a naïve strain of *A. aegypti*. This leaves the possibility there has been insertion of viral sequences in the strain where small RNAs were sequenced. The authors need to show that this is not a case to make claims about the origin of the lao ning-derived piRNAs.

As we showed in Figure 2 in the first manuscript and now in Figure EV2 in the revised manuscript, piRNAs mapping to the Liao ning virus segment 5 do not map to the *Aedes aegypti* reference genome *Aaeg* L5.0. This observation suggests that fragments of the Liao ning virus genome have been inserted into the genome of a sub-population of *Aedes aegypti*, including the strain we used in our study yet excluding the Liverpool strain used for *Aaeg* L5.0 genome. As Reviewer#2 pointed out, a strong viral infection triggers piRNA production from both sense and antisense

viral strands (Ma et al, PMID: 33419731). However, our data showed that piRNAs are exclusively produced in the antisense strand of segment 5 (Figure 2E), and barely any piRNAs were detected from other segments (Figure 2F). This pattern of piRNA production specifically from the antisense fragments of segment 5 is consistent with the idea that only these fragments were stably integrated into the genome to produce piRNAs.

We modified the results section (lines 133 to 145) and the methods to include these additional analyses.

- The authors should mine public databases for all similar datasets to expand the reach of the conclusions of the article.

As the reviewer pointed out, several previous studies have characterised somatic piRNAs in insects. However, our study specifically looked at their expression in the ovarian somatic tissue where *envelope*-carrying *gypsy* elements are known to be active in *Drosophila*. Since our primary focus is the relationship of the piRNA defence and these infectious retrotransposons and our study is the first study to investigate this outside *Drosophila*, we did not analyse data from previous studies for comparisons.

- The phylogeny shown in Fig 6A would benefit from adding more orthologs from species across insecta. This would better illustrate the relationships of the PIWI proteins of interest. The phylogeny should also include Ago1 orthologs to reinforce the identity of the genes of interest are in fact PIWIs.

In Figure 5A of the revised manuscript, we included AGO1 orthologs and several other insect Piwi/Aubergine homologs to better capture the diversity and the identity of the insect Piwi/Aubergine proteins.

Minor comments

- No description of phasing analysis in methods

We now included the description in the methods.

- The authors may consider consolidating figures and moving genome density plots to supplement to focus readers on most important data points.

We moved genome density plots that are conceptually redundant to the supplement and consolidated figures to make the first half of the results section more concise.

- The authors should include scientific names and common names of the species analyzed in the abstract to help readers be clear as to which species is which

We modified the text as suggested.

- The authors should add common names to figures to help reader understand the significance of data points

We modified figures as suggested wherever possible.

Referee #2:

The authors present a very interesting comparative analysis of ovary somatic cell piRNA expression across multiple insect species. By profiling small RNAs from the ovaries and eggs of mosquitoes, bees, crickets, and *Drosophila*, and then focusing on just the ovary-enriched piRNAs that could represent the somatic compartment, the authors argue that somatic piRNA defenses against gypsy retrotransposons have been maintained for over 400 million years, while diverging in their biogenesis mechanisms. This work provides valuable evolutionary insight into small RNA-based genome defense and its potential relationship to antiviral responses. The study is novel though it would benefit from some revisions to the issues below to strengthen mechanistic evidence and enhance quantitative analysis.

1. Authors description of somatic piRNA abundance from the enrichment method needs additional quantification and description of cutoffs to indicate how deep is the capture of somatic piRNAs since this analysis method relies on subtracting egg piRNAs from ovary piRNAs to use enrichment to represent somatic follicle cell-contributing piRNAs. This enrichment is easy to observe in the scatterplots of all the other insects except for the bee *T. carbonaria*, where piRNAs enriched in the ovaries is more murky and indistinct compared to the other insects where all the ovary enriched piRNAs are quite distinct. This issue is relevant to the next point because if the ovary-specific enrichment in bee ovaries is not clear, it could be likely a lot more germline piRNAs greatly influenced by ping-pong is dominating the signal of ping pong in Fig. 6 for bees.

In the revised manuscript Figure EV4, we measured the fractions of cluster-derived piRNAs in ovarian and embryonic small RNA libraries from all species, showing their enrichment and the abundance in the ovaries compared to the embryos. Cluster-derived piRNAs showed about 100-fold enrichment in the ovaries for species that showed great separations between the ovarian and embryonic piRNA pools, while

Bactrocera trioni, *Anopheles stephensi* and *Tetragonula carbonaria* showed less striking separations. However, our analysis showed that the cluster-derived piRNAs make up more than 30% of the *T. carbonaria* ovarian piRNA pool, yet less than 3% of the embryonic piRNA pool, suggesting that the majority of cluster-derived piRNAs come from the somatic tissue. Based on this observation and the striking ping-pong signature (e.g. the 1U and 10A enrichment is nearly 90%), we conclude that ping-pong piRNA biogenesis takes place in the ovarian somatic cells, although we do not rule out that the same cluster also produce piRNAs in the germline.

We added a sentence in the results section (lines 237 to 241) to explain this point.

2. The assessment in Fig 6 that the likely somatic compartment of the bee is just ping pong and not at all any phasing is not completely convincing by what is presented in Fig. 6, since there is just the Fig 6 D-F, but where is the negative evidence of no phasing at all? We do not see a phasing analysis diagram like what has been shown in Fig. 2 of Gainetdinov et al PMID: 30193099 or Fig. 7 of Ma et al PMID: 33419731 that would rule out phasing. It would also be helpful to readers to include in parentheses by the insect species name the colloquial name for at least one panel when there are many different species being compared. For example in Fig. 6 , *A. domesticus* (house cricket), *T. carbonaria* (stingless bee), *T. oceanicus* (field cricket). If the authors cannot convincingly strengthen the claim of no somatic piRNA phasing in bees, the conclusions in the discussion around somatic piRNA biogenesis in bees in the Discussion should also be tempered.

The phasing analysis equivalent to Gainetdinov et al is now included in Figure 6 of the revised manuscript. We also expanded the method section accordingly. The analysis showed that phasing signature is only visible in mosquito cluster piRNAs. As Gainetdinov et al and their previous studies showed, this may reflect the fact that piRNAs are trimmed from their 3' ends, hence obscuring the linkage. We cannot rule out or test this possibility without knowing the identity of the potential exonucleases. We included this point in the discussion (lines 315 to 320).

We added the common names of the insect species in the figures wherever possible.

3. The authors need to provide better normalized quantitative of the reads per million mapping to gypsy versus other piRNA clusters as a fraction of all small RNAs being the enriched fraction of the ovary versus the egg when comparing pathway strength among species. This will strengthen the claim of conserved piRNA defense rather than simply shared presence.

In Figure EV4C of the revised manuscript, we included a quantification of sense and antisense *gypsy* piRNAs from all genome mappers and specifically from the ovarian somatic clusters as a fraction of total piRNAs in each category. For all species studied, antisense *gypsy* piRNAs are more abundantly expressed than sense

piRNAs, and the fraction of antisense *gypsy* piRNAs is greater in the ovarian somatic clusters compared to the total genome mappers. These observations suggest a conserved and specialised anti-*gypsy* piRNA defence in insect ovarian somatic cells.

We expanded the results section (lines 183 to 199) to explain this new result.

4. The rendition of genomic strand piRNA mapping and Sense/Antisense orientation in the Figures of 1D and Fig. 2B and Fig. 3B, 3B' and 3B' and Fig. 4B and Fig. 5B, 5B' and 5C and Fig. S1, S2, could be improved since this is salient to the description of the other insect piRNA clusters being similar to the Dmel Flamenco piRNA cluster. The current state of the figure is to present "plus- and minus-strands are coloured in dark and light gray, respectively" which is challenging to discern. Authors should follow the convention of red and blue for plus and minus strand read mapping to allow readers to discern the polarity bias. Also, arrows for the *gypsy* transposon elements would help delineate if the TEs are mapping in plus or minus strand orientation. A much better arrangement of this figure would help the reader see if the conclusion is true that the other insect piRNA are primarily antisense to the *gypsy* elements.

As the reviewer pointed out, it is challenging to make the genomic piRNA density plot accessible as it includes many different information. We prioritise the relationship between the orientation of the *gypsy* fragments and the piRNA expression. Since piRNAs are mostly expressed from single strands, we kept the dark gray/light gray distinction as they are, whilst we changed the colour of the antisense *gypsy* fragments to red to make them more visible. Sense/antisense distinction of *gypsy* piRNAs is also measured globally in Figure EV4.

5. The mapping of the aedes aegypti ovarian piRNAs to Liao Ning Virus (LNV) is interesting, but the Fig. 2E, 2F and 2G and unclear how significant are these mappings to indicate that their Ae aegypti strain is being truly infected by LNV or if instead their Australian strain of Ae. aegypti just has taken up segment 5 of LNV into its genome as a potential new Endogenous Viral Element (EVE). We recommend the authors see the study by Ma et al PMID: 33419731 where a strong insect virus infection in mosquitoes should generate viral piRNA through the plus and minus mapping reads of the virus genome, whereas this example is only to segment 5 but not any of the other 11 segments of the LNV genome? Also, the vast majority of these LNV viral piRNAs are just AntiSense, and the authors should examine and see also a strong bias for AS piRNAs to the nrEVEs. When the authors have properly compared nrEVE piRNAs to this putative LNV-targeting piRNAs, we recommend the authors revise their text to acknowledge they do not have direct evidence of an actual LNV infection in their Ae. aegypti strain but more likely they are observing an appearance of an EVE based off of an LNV segment.

Please see the response above to Reviewer #1.

6. There should be a small RNA library sequencing statistics and mapping statistics table in the supplement showing that ovary and egg libraries are all sequenced to the equivalent and sufficient depth. If some of the small RNA libraries are sequenced to be below 10M reads those should be noted as potentially too shallow.

We now included the statistics in Figure EV4A, showing that all small RNA libraries have depth >10 million genome-mapping piRNA reads.

7. Authors view of "shared origin of antiviral and anti-retrotransposon defenses in *Aedes aegypti*" is intriguing but not directly tested. This claim needs to be supported by additional evidence for overlap or shared expression of clusters that produce both viral- and gypsy-targeting piRNAs, otherwise authors need to tone down the claim to "possible shared origin."

In the revised Figure 2B, we included the fraction of sense and antisense viral fragments (nrEVEs) in individual *Aedes aegypti* clusters along with gypsy and BEL insertions. The data show that several clusters abundantly contain both antisense nrEVEs and gypsy, suggesting their shared expression.

8. To support the authors claim of "long-term arms race" hypothesis, they need to include evidence of current or recent TE activity, such as polymorphic insertions or TE expression profiles. Without demonstrating ongoing mobility, the evolutionary narrative remains speculative.

In Figure EV5 of the revised manuscript we investigated genomic gypsy insertions that share homology (>80%) to the gypsy fragments present in the ovarian somatic piRNA clusters. For the two mosquito species, stingless bees and the two cricket species, we identified such genomic gypsy elements with GAG, POL and ENV open reading frames, indicating their current or recent activities, which are likely to be under control of ovarian somatic piRNAs.

We expanded the results section (lines 183 to 199) to cover this analysis.

9. The paper says "The coordinates of the piRNA clusters used in this study can be found in the git repository." But this table of piRNA cluster coordinates should really be part of this paper, not just in the git repository.

We now included this information in the supplementary Table S2.

Dear Dr. Hayashi

Thank you for the submission of your revised manuscript to our offices. We have now received the enclosed reports from the referees that were asked to assess it. EMBOR-2025-62715V2 still has minor comments that I would like you to address before we can proceed with the official acceptance of your manuscript. Please refer to those in a point-by-point response letter.

Furthermore, our editorial assistants' have made some more technical comments that must be resolved before your manuscript can be accepted. Please note you need to fix these issues, but you do not need to address them in the point-by-point letter.

Yehu Moran
Academic Editor
EMBO Reports

****Editorial Assistants' comments****

Data Availability Statement: in, but needs to be placed before Acknowledgments

COI/DCIS: in, but it needs to be renamed to Disclosure and Competing Interests Statement

AUTHORS: last name discrepancy - Patricia E Correira in the manuscript text vs. Patricia E Correira in the system; Should be fixed.

AFFILIATIONS: is affiliation #7 a biotech company? employment in a biotech company should be stated in DCIS.

AC/CRedit: needs to be removed from the manuscript text and appear only in the submission system.

FIGURE CALLOUTS: callout for Figure 1C appears to be missing or it's either in 'Figures EV1B, C, E and F' - needs to be called out more clearly; missing a callout for Fig. 2E; Supplementary Figure EV2C is not a correct callout.

DATASET EV LEGENDS: 2 supp. tables provided, but the nomenclature is not correct; each needs to be updated to a dataset, so that we have Dataset EV1, Dataset EV2 in all places; the legends need to be removed from the manuscript text and each should be provided in its corresponding Excel file in a separate sheet.

SYNOPSIS IMAGE: missing, please provide.

SYNOPSIS TEXT: missing, please provide.

R&T TABLE: missing, please provide.

SOURCE DATA (SD): SD provided with completed SD checklist, but there are also 2 'supplement' PDFs uploaded separately, these need to be part their corresponding Figure folders and subfolders.

Extra NOTES:

- Materials and methods should be renamed Methods

- Figure Legends need to go at the very end of the manuscript text, after the References

- The nomenclature in EV Figure files and EV figure legends is not OK: it should be Figure EV1, etc. instead of Expanded View Figure EV1 f(or Figure 1)

*Please note that the specific URL for GSE305623 dataset is not provided in the data availability statement. Please correct.

Figure Legends - Comments

- Please note that information related to n is missing in the legends of figures 6C', 7C, D; EV6 D. Please provide.

****Referee comments****

Referee #1:

I thank the authors for their responses to my recommendations and changes made to the manuscript. However, I remain concerned about the strength of the claim that phasing in *Aedes* somatic ovarian cells is slicing independent. Without mutation of Ago3 homologs to remove responder piRNAs it cannot be definitively stated that phasing in *Aedes* is slicing independent. Co-expression of multiple piwi proteins in *Aedes* soma should not be overlooked.

The authors should soften their claim regarding "slicer independence" in the text of the manuscript. In figure 8D the authors should consider an alternative label. Instead of "Slicer Independent" it could be "no ping pong linkage" phasing.

Other than this issue, I feel the article is suitable for publication in EMBO Reports

Referee #2:

I am satisfied with the revision to this manuscript.

Dear Yehu,

We thank reviewers for going through our revised manuscript and for their valuable comments. We agree with the Reviewer #1 that the slicing-dependent phasing in *Aedes* mosquito ovaries cannot be completely excluded in our current data. In reflection of this, we made textual changes in the abstract, the Results and the Discussion sections, and Figure 8D. Specifically, we described it as “phasing—devoid of slicing signature” instead of “slicing independent phasing” in the main text and the figure. This is to emphasise that our statement is solely based on examining the piRNA sequence signature. In the Results section we added a sentence stating that AGO3 is known to be expressed in the somatic cells of *Aedes* mosquitoes. We kept “slicing independent phasing” in the abstract, but softened the strength of the sentence itself by replacing “we reveal” with “we observe” and inserting “appears” to be more factual about the evidence that the manuscript provides. I hope these changes increase the clarity of the manuscript.

Sincerely,
Rippe

Referee #1:

I thank the authors for their responses to my recommendations and changes made to the manuscript. However, I remain concerned about the strength of the claim that phasing in *Aedes* somatic ovarian cells is slicing independent. Without mutation of Ago3 homologs to remove responder piRNAs it cannot be definitively stated that phasing in *Aedes* is slicing independent. Co-expression of multiple piwi proteins in *Aedes* soma should not be overlooked.

The authors should soften their claim regarding "slicer independence" in the text of the manuscript. In figure 8D the authors should consider an alternative label. Instead of "Slicer Independent" it could be "no ping pong linkage" phasing.

Other than this issue, I feel the article is suitable for publication in EMBO Reports

Referee #2:

I am satisfied with the revision to this manuscript.

Dr. Rippei Hayashi
The John Curtin School of Medical Research, Australian National University
Division of Genome Sciences and Cancer
131 Garran Road
Acton, Australian Capital Territory 2601
Australia

Dear Dr. Hayashi,

I am very pleased to accept your manuscript for publication in the next available issue of EMBO reports. Thank you for your contribution to our journal.

You may qualify for financial assistance for your publication charges - either via a Springer Nature fully open access agreement or an EMBO initiative. Check your eligibility: <https://link.springer.com/journal/44319/how-to-publish-with-us>

Yours sincerely,

Yehu Moran
Academic Editor
EMBO Reports

>>> Please note that it is EMBO Reports policy for the transcript of the editorial process (containing referee reports and your response letter) to be published as an online supplement to each paper. If you do NOT want this, you will need to inform the Editorial Office via email immediately. More information is available here: <https://link.springer.com/partners/embo-press/editorial-policies#Peer%20review>